EMBO
Molecular Medicine

# Diagnostic host gene signature for distinguishing enteric fever from other febrile diseases

Christoph J Blohmke[1,2,*] (iD), Julius Muller[3], Malick M Gibani[1,2], Hazel Dobinson[1,2], Sonu Shrestha[1,2], Soumya Perinparajah[1,2], Celina Jin[1,2], Harri Hughes[1,2], Luke Blackwell[1,2], Sabina Dongol[4], Abhilasha Karkey[4], Fernanda Schreiber[5], Derek Pickard[5], Buddha Basnyat[4], Gordon Dougan[5], Stephen Baker[6], Andrew J Pollard[1,2,†] & Thomas C Darton[1,2,6,7,†]

## Abstract

Misdiagnosis of enteric fever is a major global health problem, resulting in patient mismanagement, antimicrobial misuse and inaccurate disease burden estimates. Applying a machine learning algorithm to host gene expression profiles, we identified a diagnostic signature, which could distinguish culture-confirmed enteric fever cases from other febrile illnesses (area under receiver operating characteristic curve > 95%). Applying this signature to a culture-negative suspected enteric fever cohort in Nepal identified a further 12.6% as likely true cases. Our analysis highlights the power of data-driven approaches to identify host response patterns for the diagnosis of febrile illnesses. Expression signatures were validated using qPCR, highlighting their utility as PCR-based diagnostics for use in endemic settings.

**Keywords** biomarker; enteric fever; machine learning; transcriptomics
**Subject Categories** Biomarkers; Chromatin, Transcription & Genomics; Microbiology, Virology & Host Pathogen Interaction

## Introduction

Enteric fever, a disease caused by systemic infection with *Salmonella enterica* serovars Typhi or Paratyphi A, accounts for 13.5–26.9 million illness episodes worldwide each year (Buckle *et al*, 2012; Mogasale *et al*, 2014). In resource-limited tropical settings, these infections are endemic and the accurate diagnosis of patients presenting with undifferentiated fever is challenging.

Diagnostic tests for enteric fever rely on microbiological culture or detection of a serological response to infection, and are often unavailable or insufficiently sensitive and specific (Parry *et al*, 2011). Blood culture remains the reference standard against which new diagnostic tests are evaluated, and the sensitivity for this test can reach 80% under optimal conditions (Waddington *et al*, 2014), but low blood volumes and uncontrolled antibiotic use often result in decreased sensitivity in the field. It has been estimated that approximately 1–4% of suspected enteric fever cases are confirmed by positive blood culture, indicating a large portion of suspected cases treated with empirical antimicrobial therapy (Andrews *et al*, 2018). Due to the lack of reliable diagnostics, this culminates in substantial overtreatment of enteric fever with unnecessary antibiotics (Andrews *et al*, 2018). These reports highlight the urgent need of new diagnostic approaches to enable the accurate detection of enteric fever cases in endemic settings, to guide management of febrile patients and appropriate use of antimicrobials, and to identify populations likely to benefit from vaccine implementation.

Most common tests used for acute infectious disease diagnosis employ methods to directly detect the disease-causing pathogen, by either culture, antigen detection or amplification of genetic material by PCR. An alternative approach is to identify a set of human host immune responses, which together may generate a specific pattern associated with individual infections or pathogens. With an increasing quantity of molecular host response data being generated by high-throughput methods—including whole blood gene expression profiling—differences in the activation status of the immune response network during infection may be a tractable diagnostic approach. Recently, small sets containing 2–3 genes have been described, the expression of which can accurately differentiate between viral or bacterial infection, and active or latent tuberculosis (Herberg *et al*, 2016; Sweeney *et al*, 2016). Merging available

1 Department of Paediatrics, Centre for Clinical Vaccinology and Tropical Medicine, Oxford Vaccine Group, Oxford, UK
2 Oxford National Institute of Health Research Biomedical Centre, University of Oxford, Oxford, UK
3 The Jenner Institute, University of Oxford, Oxford, UK
4 Patan Academy of Healthy Sciences, Oxford University Clinical Research Unit, Kathmandu, Nepal
5 Infection Genomics Program, The Wellcome Trust Sanger Institute, Hinxton, UK
6 The Hospital for Tropical Diseases, Wellcome Trust Major Overseas Programme, Oxford University Clinical Research Unit, Ho Chi Minh City, Vietnam
7 Department of Infection, Immunity and Cardiovascular Disease, University of Sheffield, Sheffield, UK
*Corresponding author. Tel: +44 3934 88421862; E-mail: christoph.j.blohmke@gsk.com
†These authors contributed equally to this work as senior authors

well-characterized datasets derived from human clinical samples representative of a variety of fever-causing infections common in tropical settings presents an invaluable resource to identify host immune response patterns specific for enteric fever.

As a human-restricted infection, the development of enteric fever diagnostics has been hindered by the lack of reliable *in vivo* models. Using data from a series of controlled human infection models (CHIMs; Waddington *et al*, 2014; Dobinson *et al*, 2017) of *S.* Typhi or *S.* Paratyphi A infection, whole blood gene transcriptional responses were identified and then further characterized using samples collected from febrile patients in an endemic setting (Kathmandu, Nepal). Integrating these data with publically available human gene transcription datasets, we employed a machine learning algorithm to identify an expression signature that could distinguish blood culture-confirmed *enteric fever (EF)* cases in both the controlled environment (CHIM) and endemic setting from other febrile disease aetiologies and non-infected individuals (healthy controls; Data ref: Berry *et al*, 2010b, Data ref: Hoang *et al*, 2010b, Data ref: Idaghdour *et al*, 2012b, Data ref: Naim *et al*, 2011; Berry *et al*, 2010a; Hoang *et al*, 2010a; Idaghdour *et al*, 2012a; Obermeyer & Emanuel, 2016).

# Results

### Transcriptional profiles in response to enteric fever are similar in challenge study and endemic cohorts

We recently described the molecular response profile of acute enteric fever in individuals participating in the typhoid CHIM (Data ref: Blohmke *et al*, 2016b), which was characterized by innate immunity, inflammatory and interferon signalling patterns (Blohmke *et al*, 2016a).

To compare responses to enteric fever occurring during natural infection in an endemic area, we generated transcriptional profiles in samples collected from culture-confirmed enteric fever patients (*S.* Typhi: "03NP-ST"; *S.* Paratyphi A: "03NP-SPT"), healthy community controls ("03NP-CTRL") and febrile, culture-negative suspected enteric fever cases ("03NP-sEF") recruited in Nepal (Kathmandu; Study: "03NP"; Fig 1A). We detected significant differential expression (DE; FDR < 0.05, FC ± 1.25) of 4,308 and 4,501 genes in enteric fever patients with confirmed *S.* Typhi ($n = 19$) and *S.* Paratyphi A ($n = 12$) bacteraemia, respectively, when compared with healthy community controls ($n = 47$; Fig 1B). Similar numbers of genes were differentially expressed in samples collected at the time of enteric fever diagnosis in healthy adult volunteers challenged with either *S.* Typhi ("T1-ST") or *S.* Paratyphi A ("P1-SPT") in a CHIM (Fig 1B; Blohmke *et al*, 2016a; Dobinson *et al*, 2017).

As comparison of host responses at the gene level can be difficult to interpret, we performed gene set enrichment analysis (GSEA; Subramanian *et al*, 2005) of blood transcriptional modules (BTMs) as a conceptual framework to interpret the host responses in the context of biological pathways and themes (Li *et al*, 2014). Overall, between 54 and 74 BTMs were significantly enriched [Benjamini–Hochberg (BH)-corrected $P < 0.01$] in blood culture-confirmed enteric fever cases in the CHIM and natural infection and CHIM participants who did not develop enteric fever (measured at day 7 post-challenge— "nD7"; Appendix Table S1). Figure 1C shows the enriched BTMs in each population with lines indicating when specific BTMs are also

enriched in other populations (Fig 1C; please also refer to Appendix Table S1). The majority of BTMs enriched in cases from the enteric fever CHIM were also enriched in naturally infected cases from Nepal (56–69%, Appendix Table S1—red squares). Positively enriched modules represented cell cycle (CCY), type I/II interferon and innate antiviral responses (IFN), dendritic cell (DC), innate immunity, inflammation and monocyte (Infl./Mono) signatures. In contrast, T-cell (TC) signatures were down-regulated in patients with confirmed enteric fever, as we have previously described (Fig 1C–E; Blohmke *et al*, 2016a). In addition, a number of modules including inflammasome receptors (M53), monocyte enrichment (M118.0, M118.1, M81, M4.15, M23, M73, M64, S4) and inflammatory responses (M33) were significantly enriched in the CHIM but not in cases from Nepal. Single-sample GSEA (ssGSEA) demonstrated the similar enrichment pattern for a selection of IFN and DC signatures between individuals with confirmed typhoid and paratyphoid fever in the CHIM and naturally infected cases (Fig 2A). Overall, we observed marked similarity in the gene transcription responses between acute enteric fever cases from the CHIM and an endemic setting in Nepal.

### Responses of febrile, culture-negative samples in Nepal

In culture-negative, suspected enteric fever patients (03NP-sEF) from Nepal, we detected differential expression of 3,517 genes when compared with healthy community controls (Appendix Fig S1B). While we observed 2,843 genes as commonly expressed in all three Nepali patient cohorts (03NP-ST, 03NP-SPT and 03NP-sEF), additional 582, 756 and 183 genes were uniquely expressed by subjects with confirmed *S.* Typhi (03NP-ST), *S.* Paratyphi A (03NP-SPT) or suspected enteric fever (03NP-sEF), respectively (Appendix Fig S1A and B). Unsupervised hierarchical clustering of these patients based on their expression of the 500 most variable genes in the Nepal cohort demonstrated clustering into three groups (Fig 2B): Group A contained mostly healthy control participants (03NP-CTRL); Group B contained mostly patients with suspected enteric fever (03NP-sEF); and Group C contained a mixture of patients with suspected enteric fever, and blood culture-confirmed *S.* Typhi (03NP-ST) or *S.* Paratyphi A (03NP-SPT) infection. Of note, three samples (03NP-CONT) in this cohort grew bacterial contaminants and were thus excluded from the entire analysis.

Using ssGSEA, we observed a heterogeneous BTM enrichment pattern with broad variability in normalized enrichment scores across suspected enteric fever patients (depicted by the interdecile range; Appendix Fig S1C). The most consistent positively or negatively enriched modules represented cell cycle, IFN, inflammatory responses, DC and some NK cell signatures (green cluster) and TC- and BC-related signatures (red cluster), respectively. In contrast, heterogeneous enrichment in which approximately half of participant samples demonstrated up- or down-regulation was observed in BTMs representing TC activation patterns, protein folding and metabolism (brown cluster), or in innate response and monocyte signatures (purple cluster; Appendix Fig S1C).

### Multicohort data quality assessment

In order to address the potential overdiagnosis of enteric fever and associated inappropriate antimicrobial use, we next aimed to identify a set of genes whose expression is able to differentiate enteric fever from

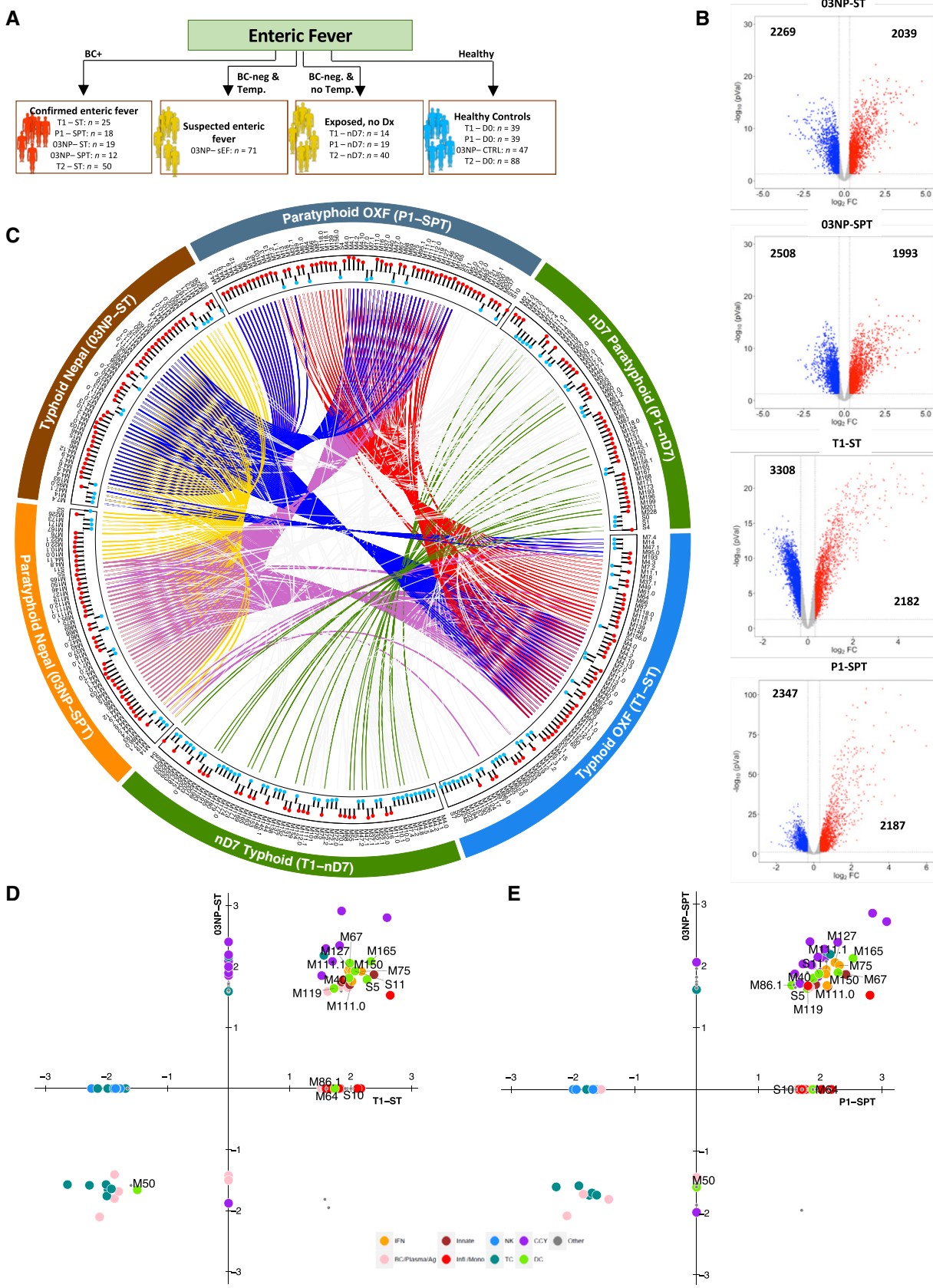

**Figure 1.**

◄

**Figure 1. Overview of Oxford and Nepal comparison.**

A   Overview of enteric fever cohorts used in this study (T1: typhoid CHIM study 1; T2: typhoid CHIM study 2; P1: paratyphoid CHIM; O3NP: Nepali cohort. ST: *S. Typhi*; SPT: *S. Paratyphi A*; sEF: suspected enteric fever; D0: day of challenge, which represents the control samples in the Oxford CHIM; CTRL, endemic community controls; nD7: day 7 after challenge in participants who stayed well in the CHIM; BC+, blood culture positive; BC−, blood culture negative; Dx, diagnosis).

B   Volcano plots of up-regulated (red) and down-regulated (blue) genes (compared to healthy control samples) in *S. Typhi*- and *S. Paratyphi A*-positive individuals (Nepal and Oxford). Black numbers indicate the up- and down-regulated genes compared to healthy controls.

C   Circular plot depicting the overlap of BTMs between enteric fever and nD7 samples from Oxford and Nepal. Tracks (from outer to inner): cohort and samples; BTM labels; direction of enrichment (blue: down; red: up; compared to healthy controls). Cords represent overlap of enrichment between given cohorts (red: overlap between P1-SPT and T1-ST; green: overlap between T1-nD7 and P1-nD7; blue: overlap of O3NP-ST with P1-SPT and T1-ST; purple: overlap of O3NP-SPT with P1-SPT and T1-ST; yellow: overlap between O3NP-SPT and O3NP-ST).

D, E   Scatter plots of BTMs enriched ($P > 0.05$) in blood culture-positive samples in Nepal (*y*-axis) versus Oxford (*x*-axis) for typhoid fever (D) and paratyphoid fever (E). For further details on BTMs, refer to reference (Chaussabel & Baldwin, 2014).

other common febrile conditions found in tropical settings. We repurposed publically available datasets (Data ref: Idaghdour *et al*, 2012b, Data ref: Subramaniam *et al*, 2015b, Data ref: Berry *et al*, 2010b, Data ref: Kaforou *et al*, 2013b, Data ref: Hoang *et al*, 2010b, Data ref: Kwissa *et al*, 2014b, Data ref: Tolfvenstam *et al*, 2011b) describing host transcriptional response in two malaria (Idaghdour *et al*, 2012a; Subramaniam *et al*, 2015a), four tuberculosis (Berry *et al*, 2010a; Kaforou *et al*, 2013a) and four dengue cohorts (Appendix Table S2; Hoang *et al*, 2010a; Tolfvenstam *et al*, 2011a; Kwissa *et al*, 2014a). Using independent datasets, we designed a discovery cohort consisting of control samples from each respective study ($n = 220$ community controls or convalescent samples, "CTRL"), 74 enteric fever ("EF"), 94 blood-stage *P. falciparum* ("bsPf"), 67 dengue ("DENV") and 54 active pulmonary tuberculosis ("PTB") cases. An independent validation cohort consisted of 109 CTRLs, 50 EF, 19 bsPf, 49 DENV and 97 PTB samples (Fig 3). Finally, a cohort of "unknown" samples was created consisting of febrile culture-negative suspected EF cases from Nepal ("sEF"), and samples collected from CHIM study participants who did not develop enteric fever after challenge at day 7 ("nD7") and their respective pre-challenge baseline samples ("D0"; Fig 3). Using principal component analysis (PCA) to assess the variability at the level of gene expression between the cohorts indicated some distinct clustering between cases (Appendix Fig S2A), for each infection, whereas no such differences were observed with the comparator CTRL samples (Appendix Fig S2B).

### Five genes sufficiently distinguish EF from other febrile infections

With these data, we aimed to build a classifier containing a minimum set of genes that could discriminate culture-confirmed enteric fever cases from individuals with other causes of fever (class: "Rest", consisting of CTRLs, DENV, PTB and bsPf; 2-class classification; Fig 3) using a Guided Regularized Random Forest (GRRF) algorithm (Deng & Runger, 2013). Genes were ranked by frequency of selection in each of 100 iterations, and applying a selection threshold of $\geq 25\%$, we identified a putative diagnostic signature containing *STAT1* (98% of iterations), *SLAMF8* (76%), *PSME2* (39%), *WARS* (37%) and *ALDH1A1* (36%; Fig 4A). With this 5-gene signature, we were able to predict which individuals in the validation cohort had enteric fever with a sensitivity and specificity of 97.1 and 88.0%, respectively (area under receiver operating characteristic curve, AUROC: 96.7%; Fig 4B, Appendix Table S3A). Of blood culture-confirmed enteric fever cases in the validation cohort, 6 of 51 were misclassified as "Rest" (i.e. classification

probability > 0.5; Fig 4C—top), and 8 of 274 samples belonging to class "Rest" were classified as enteric fever. These included six tuberculosis and one dengue case, and a pre-challenge baseline sample from a CHIM participant (Fig 3C—bottom).

To allow comparison between the different disease conditions, we quantified expression of the five genes identified in each sample using the z-score of the geometric mean of the expression values (expression score). Significant differences in expression scores were observed between the enteric fever samples and all other conditions in both the discovery (top) and the validation (bottom) cohort (Fig 4D). Of note, there were no significant differences between the scores calculated for the control samples derived from endemic areas or naïve, healthy controls from the CHIM, indicating the homogeneity of expression of these genes in healthy controls from different studies and geographical locations.

The design of discovery and validation cohorts is likely to have an impact on the diagnostic signature selected, and we therefore exchanged the validation and discovery cohort and re-ran the analysis. Although in this experiment four instead of five genes were selected (using a threshold $\geq 25\%$), most genes included were also part of the initial signature (*STAT1*, *SLAMF8*, *WARS*) and the high predictive accuracy was maintained (AUROC: 97.2%; Appendix Fig S3A and B). These results demonstrate the ability of a small number of genes for predicting true EF cases from other febrile illnesses caused by another bacterial pathogen, and of parasitic or viral origin.

### Multiclass prediction classifies three of five conditions simultaneously

Given the apparent success of small gene expression signatures in classifying two distinct groups, we sought to leverage the overall dataset and the GRRF algorithm to identify a signature that could accurately classify more than two classes simultaneously. We re-analysed the data preserving the original class labels (i.e. CTRL, bsPf, DENV, PTB and EF) and performed the iterative feature selection step using the GRRF algorithm (Fig 3—"multiclass classification"). Applying a $\geq 25\%$ selection threshold to ranked features identified seven genes (*RFX7*, *C1QB*, *ANKRD22*, *WARS*, *BATF2*, *STAT1* and *C1QC*) able to discriminate the classes (Fig 4E). Prediction of the validation cohort using this 7-gene signature indicated good sensitivity and specificity for classifying CTRL, bsPf and EF cases; however, the identification of DENV and PTB was less accurate (Fig 4F, Appendix Table S3B). Analysis of individual gene expression levels in each group indicated that *RFX7* was only up-regulated in bsPf samples, while *STAT1*, *WARS*, and *ANKRD22* and *BATF2* were all strongly up-regulated in EF.

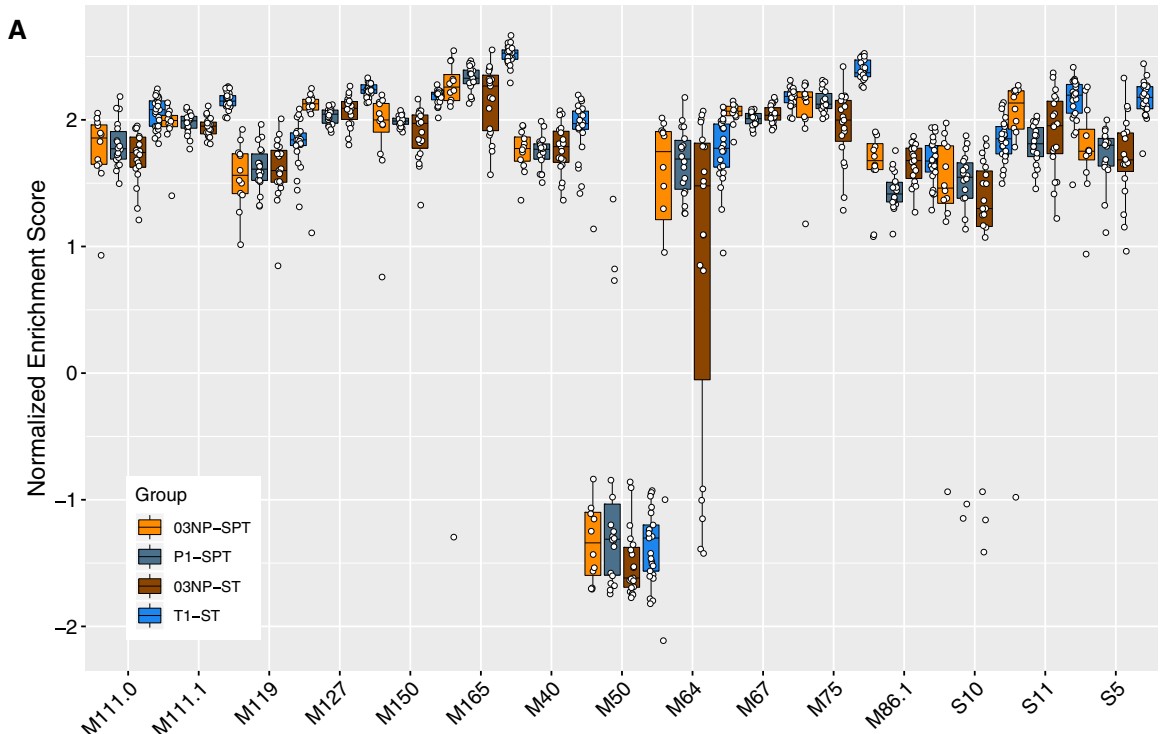

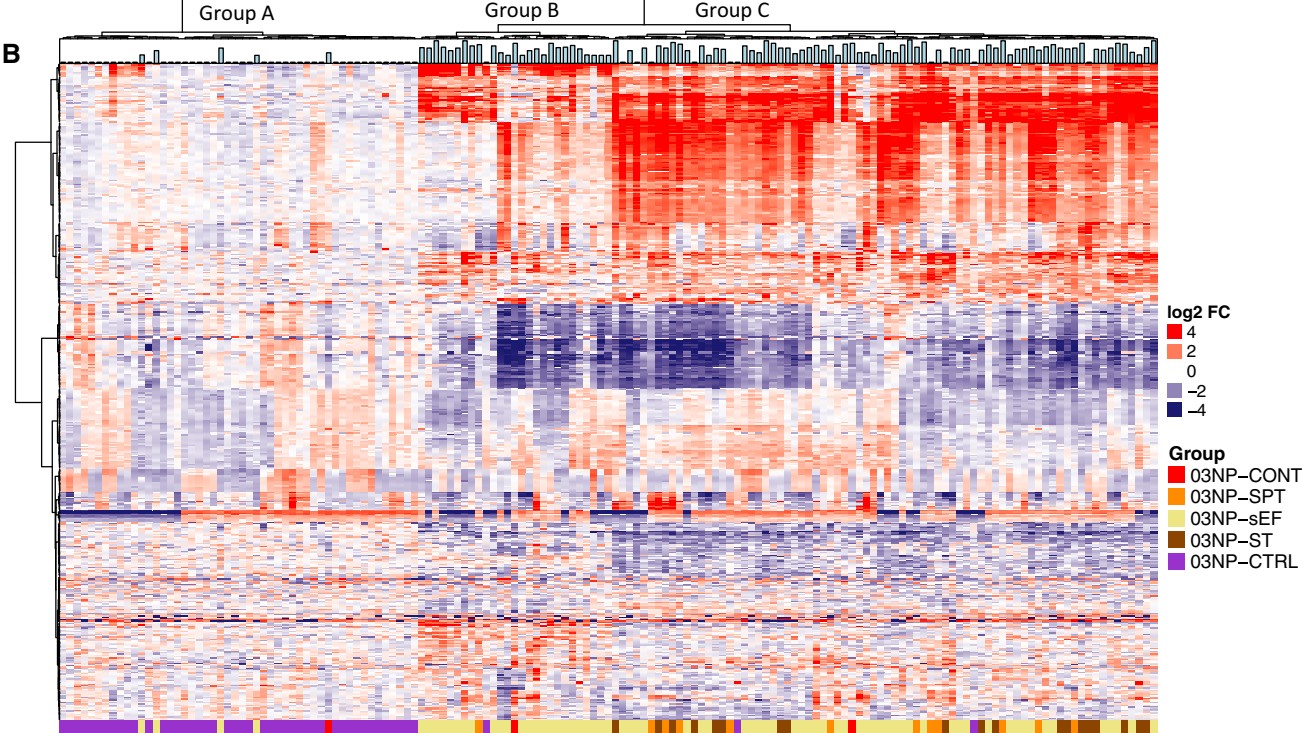

**Figure 2. BTM responses between EF cohorts and heatmap view of samples collected in Nepal.**

A Single-sample GSEA normalised enrichment scores (NES) of IFN and DC BTMs of individuals with blood culture-confirmed enteric fever in Nepal and Oxford. Data are median and 25th/75th percentile. For numbers in each group please refer to Figure 1A.

B Heatmap of the 500 most variably expressed genes in samples of the Nepali cohort. Bar graph on top of the heatmap shows temperature of each individual at the time of sampling. Three samples labelled as "03NP-CONT" are samples that grew bacterial contaminants and were thus excluded from the entire analysis.

Expression of these genes in PTB and DENV samples was variable accounting for the lower performance of the signature in these conditions (Appendix Fig S4A and B).

## Prediction of unknown samples

Given the superior performance of the 2-class diagnostic signature, our subsequent analyses focused on using the initial five genes identified to ascertain whether enteric fever was the likely true underlying aetiology of suspected febrile, blood culture-negative cases in Nepal (03NP-sEF; $n = 71$), part of the unknown cohort (Fig 3). Included in this cohort were 144 samples originating from challenge studies with known class membership confirming the correct classification of 94.4% of the samples by the GRRF algorithm (Appendix Fig S5 and Table S4).

Classification of the 03NP-sEF cases predicted nine of 71 (12.6%) febrile, culture-negative patients to be true enteric fever cases and the remaining samples to belong to class "Rest" (Fig 5A). Relating the gene expression scores to the predicted class probabilities indicated no clear separation of scores according to the predicted class (Fig 5B). Furthermore, comparing the expression score (based on microarray data) of febrile, culture-negative samples (03NP-sEF) with culture-confirmed enteric fever (03NP-ST or 03NP-SPT) in Nepal showed a marked overlap, indicating that these scores alone are insufficient for 2-class discrimination (Fig 5C).

## Diagnostic validation by qPCR

Finally, to validate the induction of the diagnostic gene signature in blood culture-confirmed enteric fever cases, we performed

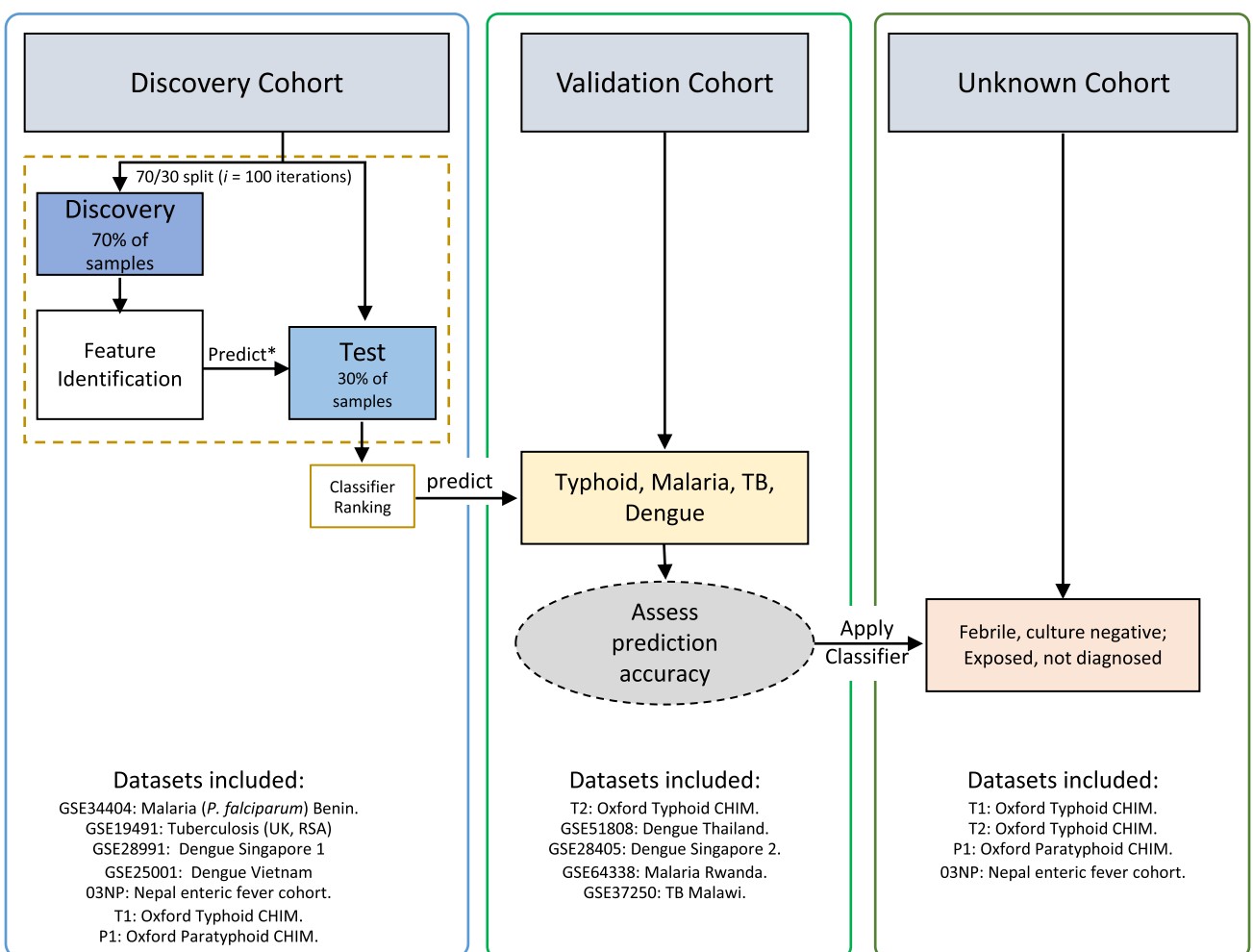

**Figure 3.  Flow diagram of machine learning analysis.**

The discover cohort consisted of only Illumina datasets and was used for feature selection using the GRRF algorithm. For the validation cohort, Affymetrix datasets were also included. A cohort of unknown samples consisted of pre-challenge baseline samples of participants who stayed well following challenge, their respective nD7 samples (7 days after challenge) and febrile, culture-negative suspected enteric fever (sEF) cases from Nepal. Refer to Appendix Table S2 for study identifiers. 03NP: Nepali cohort. T1: Oxford typhoid CHIM study 1. T2: Oxford typhoid CHIM study 2; P1: Oxford paratyphoid CHIM.

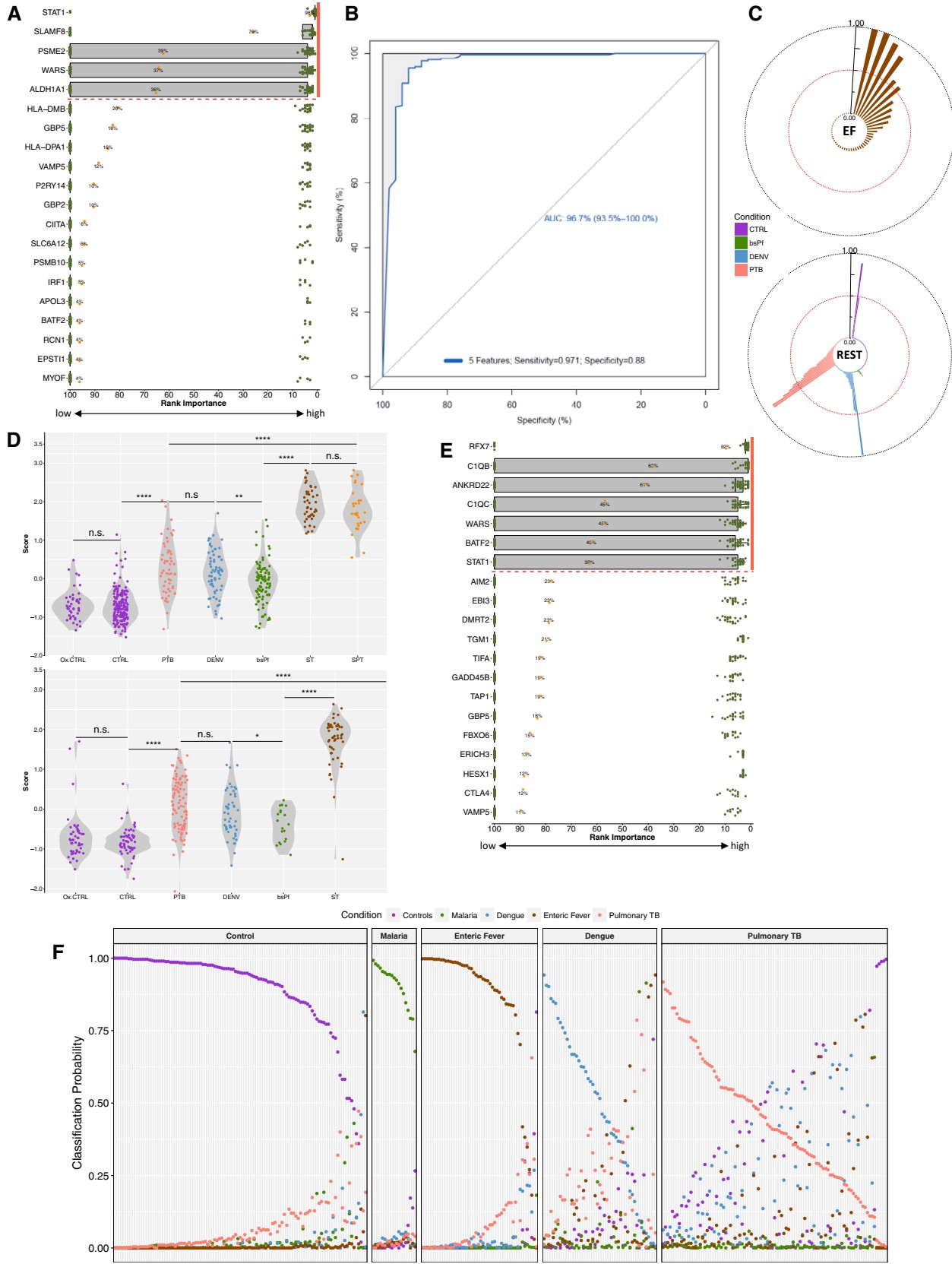

**Figure 4.**

**Figure 4. Identification of diagnostic signatures.**

A Ranking of genes by their selection frequency into the diagnostic signature out of 100 iterations (orange dot) during the 2-class classification. *Y*-axis = genes ranked by selection frequency. *X*-axis = importance measure of each gene across all 100 iterations. Green dots: importance measure for each gene per iteration. A cut-off of 25% was selected to detect a 5-gene putative diagnostic signature (orange bar).

B Performance of the 5-gene classifier when predicting the class membership of the validation cohort.

C Top: probability of an EF sample to be classified as non-EF (> 0.5). Bottom: probability of sample belonging to "Rest" to be classified as EF (> 0.5). Red dotted line signifies the 0.5 prediction probability. *Y*-axis: prediction probability ranging from 0 to 1.

D Combined expression score for samples based on the 5-gene signature for samples in the discovery cohort (top) and validation cohort (bottom). Ox.CTRL, Oxford controls (DO); CTRL, Nepali control samples; PTB, pulmonary TB; DENV, dengue samples; bsPf, blood-stage *P. falciparum*; SPT, *S.* Paratyphi A; ST, *S.* Typhi. ST and SPT samples are derived from the challenge models as well as from Nepal. Significance levels were determined using Student's *t*-test (two-sided): *$P < 0.05$; **$P < 0.01$; ****$P < 0.0001$. Number of samples per group: Discovery: Ox.CTRL = 45; CTRL = 175; PTB = 54; DENV = 67; bsPf = 94; ST = 44; SPT = 30. Validation: Ox.CTRL = 50; CTRL = 59; PTB = 97; DENV = 49; bsPf = 19; ST = 50.

E Ranking of genes by their selection frequency into the diagnostic signature out of 100 iterations during the multiclass classification. A cut-off of 25% was selected to detect a 7-gene putative diagnostic signature (orange bar).

F Classification probabilities for each sample of the validation cohort based on the 7-gene signature.

high-throughput qPCR in samples collected during an independent typhoid CHIM (Appendix Table S2—qPCR; Jin *et al*, 2017) and in the Nepali cohort. Transcription of the 5-gene signature was increased at the time of diagnosis in most participants with culture-confirmed enteric fever in both sample sets (Fig 5D and E). Two CHIM participants diagnosed with typhoid infection and one patient infected with *S.* Paratyphi A in Nepal showed low expression of all genes and a resulting low expression score (Fig 5E—black arrows). In contrast, 1 day-7 sample from a participant not diagnosed with enteric fever demonstrated high expression of the putative diagnostic gene signature (Fig 5E—black arrows).

As surrogate disease severity markers, temperature showed a poor correlation with the expression score in both CHIM and endemic setting culture-confirmed enteric fever cases (Fig 5F and G—left). In contrast, C-reactive protein levels (only available for CHIM participants) were significantly associated with the expression score of the 5-gene signature (Fig 5F and G—right), thus underlining the relevance of this signature in reflecting the clinical presentation of enteric fever. In the Nepal cohort, gene expression also strongly correlated between the array and qPCR data (Appendix Fig S6). Overall, these results verify the strong expression of the putative diagnostic signatures in samples from patients with acute enteric fever and underline the clinical plausibility through association with disease severity parameters.

## Discussion

New approaches to diagnose patients with enteric fever are urgently needed, as currently available methods are antiquated and unreliable. New diagnostic modalities are required, to both improve the immediate management of patients and increase the accuracy of disease burden measurements to support targeted vaccine implementation. Here, we demonstrate a reproducible host expression signature of five genes (*STAT1*, *SLAMF8*, *PSME2*, *WARS* and *ALDH1A1*) able to discriminate EF cases from other common causes of fever in the tropics with an accuracy of > 96% (AUROC; sensitivity 97%, specificity 88%). Application of high-throughput methods such as functional genomics, to this major health concern (Escadafal *et al*, 2017), underscores the importance and tangible benefits of applying "omics technologies" to combating infectious diseases in the most needy populations (Baker, 2011). While further optimization work is required, validating the expression of our signature

using conventional methods such as qPCR demonstrates feasibility for further development into a routine laboratory test (Jiang *et al*, 2014). A rapid, PCR-based test would be useful tool for accelerated diagnosis of enteric fever in hard-to-diagnose settings. While cost-effective, early approaches for easy use in resource-limited settings exist (Jiang *et al*, 2014; Huang *et al*, 2018), further development of reliable PCR diagnostic is needed to make such a test fit for purpose.

The degree of perturbation of molecular responses occurring during enteric fever can be confounded by the duration of clinical illness (ranging in 12 h to ≥ 3 days in the CHIM and patients from Nepal, respectively) or the specific pathogen (*S.* Typhi or *S.* Paratyphi A). This may hinder identification of a reproducible gene expression signature reliably expressed in various settings. The responses to *S.* Typhi and *S.* Paratyphi A cases in Nepal were remarkably similar despite longer duration of disease and "uncontrolled" exposure to antimicrobials in Nepal, with the majority of DE genes overlapping between the two groups, which is unsurprising given the close genetic relatedness of both pathogens (McClelland *et al*, 2004). Enrichment of BTMs resembled responses described previously by us (Blohmke *et al*, 2016a; Salerno-Goncalves *et al*, 2017) and underlined the concordance between culture-confirmed enteric fever cases from Oxford and Nepal despite the possible differences between challenge and currently circulating strains.

Despite the multiple redundancies incorporated into human immune pathways driven by successful evolution (Nish & Medzhitov, 2011), our data suggest that the pattern of immune response activation is sufficiently specific to allow identification of the causative pathogen. For example, while immune responses during enteric fever and TB are broadly characterized by IFN signalling, we and others have reported that this response during acute *S.* Typhi infection appears to be skewed towards a type II pattern likely associated with neutrophils and NK cells rather than the type I-dominated profile found in TB (Manca *et al*, 2005; Thompson *et al*, 2009; Berry *et al*, 2010a; Spees *et al*, 2014; Blohmke *et al*, 2016a, 2017; Dobinson *et al*, 2017). Application of computational methods to large datasets including host gene expression has been shown to be an effective approach to capture such differential activation of immune pathways (Herberg *et al*, 2016; Sweeney *et al*, 2016). Two of the genes identified in our 5-gene diagnostic signature are important entities in the IFN-γ signalling cascade (*STAT1* and *WARS*), which has been broadly implicated in the responses to enteric fever, TB (Berry *et al*, 2010a), dengue (De La Cruz Hernandez *et al*, 2014) and *P. falciparum* (Miller *et al*, 2014) infection. The discriminatory

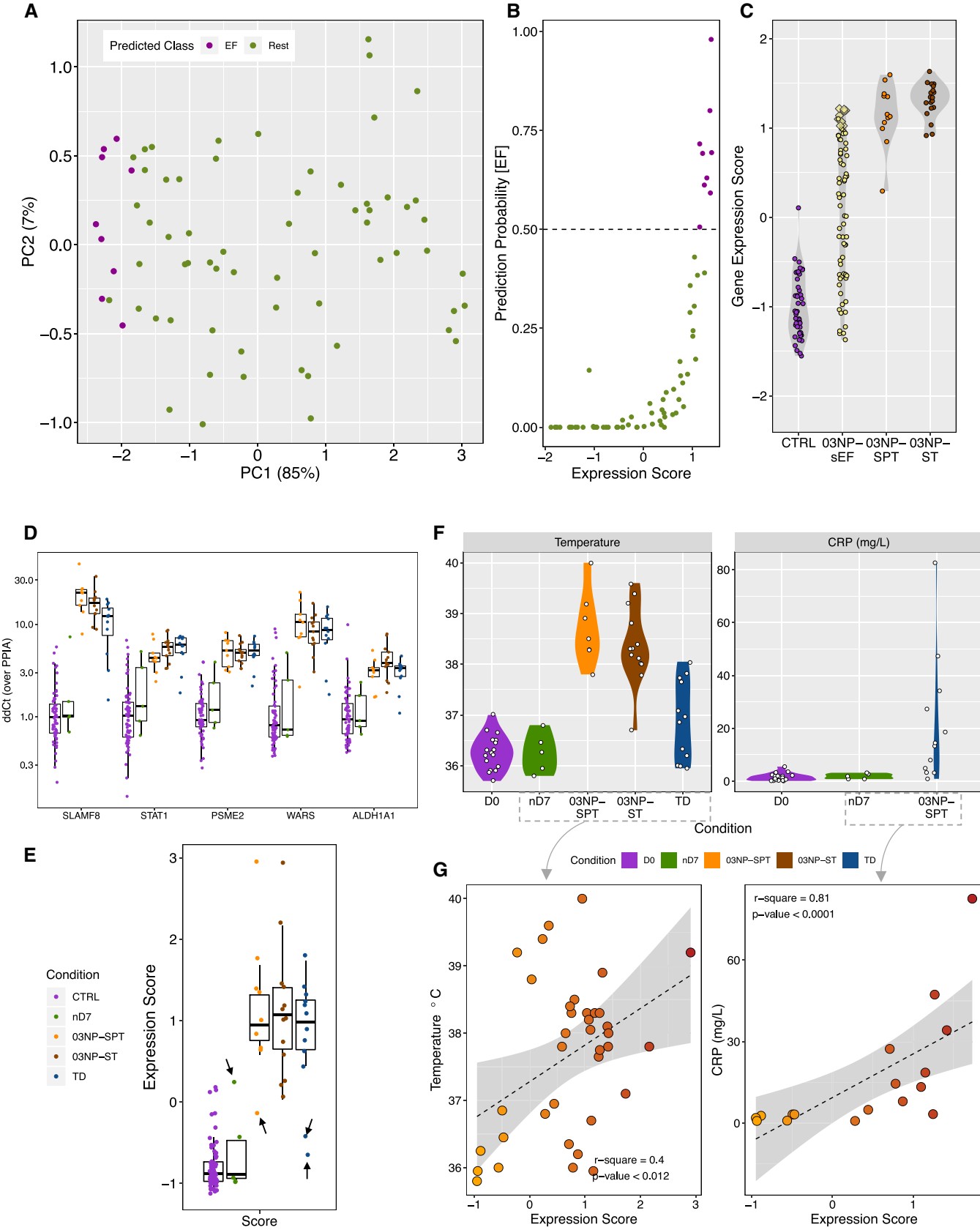

**Figure 5.**

◀

**Figure 5.  Prediction of Nepali unknown samples using the 2-class and qPCR validation.**

A  PCA of sEF samples based on the 5-gene signature (based on gene array data) coloured by predicted class membership (EF: purple; green: rest).

B  Dot plot of prediction probability of being class EF versus the expression score calculated on the bases of the 5-gene signature (based on gene array data).

C  qPCR gene expression scores of the 5-gene signature ($\Delta\Delta C_T$ over PPIA) for CTRLs, 03NP-sEF, 03NP-SPT and 03NP-ST samples from Nepal. Yellow diamonds in the 03NP-sEF category represent the nine patients classified as EF based on the Random Forest algorithm.

D  qPCR expression values ($\Delta\Delta C_t$ over PPIA) of the 5-gene signature in control samples (Oxford and Nepal), *S.* Paratyphi A (03NP-SPT) or *S.* Typhi (03NP-ST) in Nepal, samples at day 7 after challenge of participants who stayed well following challenge with *S.* Typhi (nD7), or typhoid diagnosis after challenge (TD) in the Vi-TCV study (Appendix Table S2). Colour legend in panel (E). Data are median with the $25^{th}$/$75^{th}$ percentile. *N* per group: CTRL = 64; nD7 = 5; 03NP-SPT = 9; 03NP-ST = 13; TD = 12.

E  Combined qPCR expression score of the 5-gene signature. Black arrows indicate outlier samples. Data are median with the $25^{th}$/$75^{th}$ percentile. *N* per group: CTRL = 64; nD7 = 5; 03NP-SPT = 9; 03NP-ST = 13; TD = 12.

F  Temperature and CRP for samples of which data were available (CRP was only measured in the Oxford CHIM). D0, pre-challenge baseline Vi-TCV study; nD7, day-7 samples of participants who stayed well following challenge (Vi-TCV study); SPT, *S.* Paratyphi A (03NP); ST, *S.* Typhi (03NP); TD, typhoid diagnosis (Vi-TCV study).

G  Spearman's rank correlation of the 5-gene combined expression score and (left) temperature (only nD7 and TD samples from the Oxford CHIM—Vi-TCV and SPT and ST cases from Nepal at presentation to hospital were included) and (right) CRP (CRP was only available for Oxford CHIM—Vi-TCV samples, and we excluded D0 baseline measures).

impact of increased expression of these genes identified in our analysis, however, suggests that there are distinct differences during the responses to these very different pathogens sufficient to discriminate underlying disease aetiology (Munoz-Jordan *et al*, 2003; Obiero *et al*, 2015) possibly based on subtle metabolic differences (Zhang *et al*, 2013; Blohmke *et al*, 2016a). While STAT1 and WARS are markers of an IFN-γ response, SLAMF8 is a surface-expressed protein (van Driel *et al*, 2016) found on macrophages, DCs and neutrophils and induced by IFN-γ or Gram-negative bacteria (Wang *et al*, 2012). SLAMF8 negatively regulates ROS production through inhibition of NADPH oxidase 2 (NOX2) in the bacterial phagosome and reduces ROS-induced inflammatory cell migration (Wang *et al*, 2015). While oxidative stress is a common response to infection, *Salmonella* survival is reduced in SLAMF1-deficient mice and can interfere with localization of functional NOX2 in *Salmonella*-containing vacuoles (SCVs), linking SLAM proteins and oxidative stress (Fang, 2011). PSME2 is one of the two interferon-inducible subunits of the 20S immunoproteasome (IP) regulator 11S and is involved in immune responses and antigen processing (Nandi *et al*, 1996). The 20S IP can be induced by oxidative stress and preferentially hydrolyses non-ubiquitinated proteins (Dubiel *et al*, 1992; Seifert *et al*, 2010). Thus, genes involved in these processes may be exploited to distinguish between pathogens inducing oxidative stress and those also triggering ubiquitination (Cirillo *et al*, 2009; Spooner & Yilmaz, 2011). While ALDH1A1 has not specifically been linked with responses to invasive bacterial infections, it is involved in gut-homing of TCs through expression of retinoic acid (Molotkov & Duester, 2003; Iwata *et al*, 2004), a phenotype we have observed following infection with *S.* Typhi (Salerno-Goncalves *et al*, 2017). C1QB and C1QC are well-known subunits of the complement subcomponent C1q and, together with ANKRD22 [involved in cell cycle control (Yin *et al*, 2017)], have previously been described as part of a signature able to distinguish active from latent TB (Kaforou *et al*, 2013a). The function of the transcription factor *RFX7* is largely unknown but has been found to be strongly up-regulated during blood-stage malaria, and its selection in our 7-gene signature is therefore likely to be driving the classification of malaria cases.

Of note, while multiclass classification is difficult to perform and here merely serves as demonstration that data-driven approaches may be capable of performing this task, it is interesting to observe increased misclassification rates specifically in the DENV and TB groups. In the validation cohort, the majority of misclassified DENV cases were identified as enteric fever (5/49) or TB (9/49), and

misclassified TB samples as enteric fever (13/97) or DENV (23/97), possibly reflecting the overlapping immune response seen due to the intracellular nature of all three pathogens. In the TB group, 15 of 97 samples were misclassified as controls, compared with one DENV sample being misclassified as such, for example, potentially owing to the broad clinical phenotype or lack of inflammatory/immune responses seen in the peripheral blood during tissue-specific pulmonary TB infection.

Overall, the genes identified in both signatures through our unbiased selection approach are supported by previous studies including those aiming to develop predictive diagnostic signatures (Berry *et al*, 2010a; Kaforou *et al*, 2013a; Zak *et al*, 2016). In the era of biological "big data", several studies have explored the utility of gene transcription signatures capable of discriminating viral aetiologies, viral or bacterial infections and acute or latent tuberculosis (Zaas *et al*, 2009; Kaforou *et al*, 2013a; Anderson *et al*, 2014; Andres-Terre *et al*, 2015; Herberg *et al*, 2016; Mahajan *et al*, 2016; Sweeney *et al*, 2016). Only in the tuberculosis studies have such signatures been identified from samples collected in high-incidence, disease-endemic settings and been further validated against other disease processes including (but not limited to) pneumonia, sepsis, and streptococcal and staphylococcal infections (Berry *et al*, 2010a; Anderson *et al*, 2014; Sweeney *et al*, 2016). Herberg *et al* (2016) demonstrated that distinction between viral and bacterial infections could be achieved based on two genes only. In contrast, most efforts undertaken to diagnose active TB employ biomarker signatures ranging in size from 3 to 86 genes, possibly due to broad and heterologous molecular responses seen in response to differing clinical phenotypes of infection. In our analysis, we specifically focused on pathogens with the potential to cause undifferentiated febrile illnesses in tropical settings. While the clinical presentation and epidemiology of the infections chosen may be sufficient to distinguish the aetiologies clinically, enteric fever has a broad differential diagnosis and is frequently overdiagnosed in the absence of confirmatory laboratory results. Notably, despite the high prediction accuracy of the signatures identified in our analysis, this type of data modelling is highly dependent on the quality and availability of suitable input datasets. Although an increasing amount of data is accumulating in the public domain, few well-defined datasets of samples representing a larger repertoire of febrile illnesses are available. For example, rickettsial infection is likely to underlie a large burden of the culture-negative cases in Nepal; however, no gene expression datasets exist, and the lack of adequate confirmatory diagnostic tests

further hinders the inclusion of such data in our analysis. Indirect evidence for a likely high burden of rickettsial infection was demonstrated in a recent randomized controlled treatment trial of treatment for typhoid at the same centre in Nepal (Arjyal *et al*, 2016). In this study, a higher proportion of culture-negative cases clinically responded to fluoroquinolone class antibiotics rather than the 3rd-generation cephalosporin used in the study, probably due to the high frequency of murine and scrub typhus presenting as acute undifferentiated febrile infection in this population. Of note, we have shown that non-typhoidal *Salmonella* infection only accounts for 0.6% bacteraemia over a 23-year period (Zellweger *et al*, 2018).

Although the 5-gene signature achieved high accuracy in identifying enteric fever cases, several culture-confirmed cases were misclassified. Metadata from samples collected in the Oxford CHIM indicate that the majority of these misclassified samples had a temperature below 37°C (5/6) and were diagnosed beyond 7 days after challenge (4/6), which, in our CHIM experience, is likely to indicate a less severe disease phenotype. In contrast, six nD7 samples from the Oxford CHIM (part of the unknown cohort) classified as enteric fever did show some evidence of response based on either cytokine profile, temperature or positive stool culture findings. Because our analysis was purely data-driven and not motivated by clinical suspicion, we believe that these observations and the significant association of the gene expression scores with CRP provide sufficient evidence that these study participants had infection despite not meeting our study endpoint definitions for enteric fever. Despite all this evidence, an open question remains the effect of prior antibiotic use and duration of illness on the expression of the 5-gene signature. While our study is too small to formally assess this, the similarity of gene expression between samples derived from the CHIM (short disease duration, no antibiotic exposure) and Nepal (long disease duration and exposure to antibiotics) may suggest stability of the gene signature in the presence of these co-founding factors. Additional work is required to address these limitations by performing a multisite prospective diagnostic evaluation recruiting all patients with fever to ensure a balanced case mix and range of alternate fever-causing aetiologies (e.g. rickettsial infection).

In summary, our work demonstrates how a large gene expression dataset derived from challenge study cohorts and settings endemic for febrile infectious diseases can be exploited for diagnostic biomarker discovery. Verification of the putative diagnostic signature using qPCR in independent validation sets indicates that a diagnostic test derived from these gene expression data could be developed for deployment in resource-limited settings. The application of purely data-driven analyses to large and well-defined host–pathogen datasets derived from disease-relevant populations may enable us to develop a single, highly accurate diagnostic signature, which would allow rapid identification of the main fever-causing aetiologies from readily available biological specimens.

# Materials and Methods

## Typhoid challenge model

Samples included in the discovery cohort were collected during a typhoid dose–escalation study in which 41 healthy adult volunteers

ingested a single dose of *S.* Typhi Quailes strain following pre-treatment with 120 ml sodium bicarbonate solution (Study: T1). In this study, one of the two doses was administered: $1$–$5 \times 10^3$ ($n = 21$) and $1$–$5 \times 10^4$ ($n = 20$; Waddington *et al*, 2014). Samples used in the validation cohort were collected from a second typhoid challenge model performed as part of a vaccine efficacy study (Study: T2), in which healthy adult volunteers ingested a single dose of *S.* Typhi Quailes strain ($1$–$5 \times 10^4$, $n = 99$) 4 weeks after oral vaccination with Ty21a, M01ZH09 or placebo (Darton *et al*, 2016). Lastly, samples collected from the control arm of a further vaccine efficacy challenge study, in which participants received meningococcal ACWY-CRM conjugate vaccine (MENVEO®, GlaxoSmithKline) prior to challenge, were used for the independent qPCR validation experiment (Jin *et al*, 2017). The clinical and molecular results of these studies have been described previously (Waddington *et al*, 2014; Darton *et al*, 2016; Dobinson *et al*, 2017; Jin *et al*, 2017). In all typhoid challenge studies, participants were treated with a 2-week course of antibiotics at the time of diagnosis (fever ≥ 38°C sustained for ≥ 12 h and/or positive blood culture) or at day 14 post-challenge if diagnostic criteria were not reached. Informed consent was obtained from all subjects, and all experiments conformed to the principles set out in the WMA Declaration of Helsinki and the Department of Health and Human Services Belmont Report.

## Paratyphoid challenge model

Clinical samples for paratyphoid infection were collected during a dose–escalation study, as previously described (P1; Dobinson *et al*, 2017). Briefly, 40 healthy adult volunteers were challenged with a single oral dose of virulent *S.* Paratyphi A (strain NVGH308) bacteria, which, as before, was suspended in 30 ml sodium bicarbonate solution (17.5 mg/ml), and after pre-treatment with 120 ml sodium bicarbonate solution. Oral challenge inocula were given at one of two dose levels, low ($n = 20$; median [range] = $0.9 \times 10^3$ CFU [$0.7 \times 10^3$–$1.3 \times 10^3$]) or high ($n = 20$; median [range] = $2.4 \times 10^3$ CFU [$2.2 \times 10^3$–$2.8 \times 10^3$]). Criterion for diagnosis was either microbiological (≥ 1 positive blood culture collected after day 3) or clinical (fever ≥ 38°C sustained for ≥ 12 h). Participants were ambulatory and followed up as outpatients at least daily after challenge when safety, clinical and laboratory measurements were performed (Dobinson *et al*, 2017). Informed consent was obtained from all subjects, and all experiments conformed to the principles set out in the WMA Declaration of Helsinki and the Department of Health and Human Services Belmont Report.

## Endemic cohort

To validate the gene transcriptional signatures in a relevant patient cohort, blood samples were collected from three cohorts at Patan Hospital or the Civil Hospital both located in the Lalitpur Sub-Metropolitan City Area of Kathmandu Valley in Nepal. Firstly, blood samples were collected as part of a diagnostic study (Darton *et al*, 2017) from febrile patients presenting to hospital with ≥ 3 days of fever, with no obvious focus of infection (WHO, 2018), and diagnosed with blood culture-confirmed *S.* Typhi ($n = 19$) or *S.* Paratyphi A ($n = 12$) infection, and from febrile patients who were blood culture-negative for any pathogen ($n = 71$). Samples from a cohort of healthy control volunteers ($n = 44$) were also collected as part of

this study. Informed consent was obtained from all subjects, and all experiments conformed to the principles set out in the WMA Declaration of Helsinki and the Department of Health and Human Services Belmont Report.

### Gene expression array sample processing

In the human challenge studies (T1, T2 and P1), peripheral venous blood (3 ml) was collected in Tempus™ Blood RNA tubes (Applied Biosystems) before challenge (baseline, pre-challenge controls, "D0", $n = 166$) and at paratyphoid diagnosis ("SPT", $n = 18$) or typhoid diagnosis ("ST", $n = 75$). In those challenged but who did not develop enteric fever within 14 days of challenge, gene expression was measured at the median day of diagnosis of the diagnosed group in the appropriate studies and this day was termed "nD7" ($n = 73$). In Nepal, blood was collected when patients presented to hospital ($n = 102$) and from healthy controls ($n = 44$; Fig 1A, Appendix Table S1). Total RNA was extracted from all samples using the Tempus™ Spin RNA Isolation Kit (Life Technologies). Where applicable, 50 ng of RNA was used for hybridization onto Illumina HT-12v4 bead arrays (Illumina Inc.) at the Wellcome Trust Sanger Institute (Hinxton, UK) or the Wellcome Trust Centre for Human Genetics (Oxford, UK), and fluorescent probe intensities were captured with the GenomeStudio software (Illumina Inc.). For the paratyphoid CHIM (P1), RNA gene expression was determined using RNA sequencing. Briefly, libraries were prepared using a poly-A selection step to exclude ribosomal RNA species (read length: 75 bp paired-end) and samples were subsequently multiplexed in 95 samples/lane over 10 lanes plus one 5-plex pool run on 1 lane and sequenced using a Illumina HiSeq 200 V4.

### Data pre-processing

Paired-end reads were adapter-removed and trimmed from 75 to 65 bp using Trimmomatic v0.35 (Bolger *et al*, 2014), and only reads exceeding a mean base quality 5 within all sliding windows of 5 bp were mapped to the Gencode v25/hg38 transcriptome using STAR aligner v2.5.2b keeping only multimapped reads mapping to at most 20 locations. featureCounts(), a function from the subread set of tools v1.5.1 (Brodersen *et al*, 2010), was used to quantify reads in Gencode v25 basic gene locations with parameters -C -B -M -s 2 -p -S fr. Between-sample normalization was performed using TMM (trimmed mean of M-values) normalization as implemented in the edgeR (Robinson *et al*, 2010) package, and we used principal component analysis (PCA) as quality control step and excluded two samples, which were clear outliers due to also failing QC during the library preparation. Counts were converted into $\log_2$ counts per million (cpm) values with 0.5 prior counts to avoid taking the logarithm of zero and were then taken forward to the multicohort quality control. Illumina HT-12v4 bead array data were pre-processed by background subtraction, quantile normalization and $\log_2$-transformation using the limma package in R (Ritchie *et al*, 2015). Probes were collapsed to HUGO gene identifiers keeping only the highest expressed probe.

### Data download

Previously published whole blood transcriptional array data were downloaded from the Gene Expression Omnibus (GEO) data repository. In this study, we specifically focused on studies investigating blood-stage *Plasmodium falciparum* (bsPf; two cohorts of blood-stage, HIV-negative malaria cohorts; children and adults; Idaghdour *et al*, 2012a; Subramaniam *et al*, 2015a), acute uncomplicated dengue (DENV; four adult South-East Asian cohorts of uncomplicated dengue fever patients; Hoang *et al*, 2010a; Tolfvenstam *et al*, 2011a; Kwissa *et al*, 2014a) and active pulmonary tuberculosis (PTb; four cohorts of active, pulmonary TB HIV-negative adults from Africa and the UK; Berry *et al*, 2010a; Kaforou *et al*, 2013a), all infections which present with undifferentiated fever and are relevant to areas where enteric fever is endemic (Appendix Table S2). Raw data were downloaded from GEO using the getGEO function (Davis & Meltzer, 2007) and quantile normalization with detection *P*-values and control probes where available. Probes were collapsed to HUGO gene identifiers keeping only the highest expressed probe.

### Data processing and cohort quality control

Probe sequences on microarrays may not correspond to the most recent release of the human reference genome that was used for the RNAseq alignment. In order to mitigate this potential discrepancy, we re-annotated the probes to the Gencode v25/hg38. The new annotations were used as gene names for each probe. To avoid uninformative genes and gender bias, only probes common to all datasets, not located on sex chromosomes and with an expression above the lowest tertile of the average expression (12,821 probes), were used and a "superset" was created by merging the expression data from all studies into one large data matrix. In order to avoid platform or study-related artefacts between the data, we applied surrogate variable analysis (sva; Leek *et al*, 2017) to remove batch effects based on study ID while preserving the disease condition (i.e. control or individual infection).

### Diagnostic signature identification

For classification analyses, we separated the superset into a discovery cohort and a validation cohort. To ensure heterogeneity and optimal feature identification, we restricted the discovery cohort to samples solely generated on Illumina platforms and ensured inclusion of EF samples from Oxford and Nepal. In order to establish a validation cohort, we casted a wider net and permitted studies generated on other platforms including Affymetrix due to the limited amount of suitable datasets available in the public domain. In addition, to predict unknown samples by applying the signatures identified in this study, we separated the febrile, culture-negative suspected enteric fever cases, samples at day 7 after challenge of those who stayed well and their respective pre-challenge control samples from the superset into a cohort of samples of unknown aetiology (unknown cohort; Fig 3).

Only the discovery cohort was used for feature selection using Guided Regularized Random Forest (GRRF; Deng & Runger, 2013) as implemented in the R package RRF v1.7 (preprint: Deng, 2013) with $\gamma = 0.5$, and parameter mtry tuning was performed using the tuneRRF command. Feature selection was repeated on 100 iterations of bootstrapped subsets of about 70% of the data in the discovery cohort. To assess model performance, predictions on the remaining 30% of the discovery cohort were performed and balanced

**The paper explained**

**Problem**
Enteric fever caused by *Salmonella* Typhi and *S.* Paratyphi A causes significant morbidity in resource-limited settings. Undifferentiated febrile presentation and inadequate diagnostic tests make enteric fever difficult to diagnose, leading to missed cases or inappropriate antimicrobial use. This leads to late and/or missed diagnosis and possible overuse of antimicrobials.

**Results**
This study suggests a novel approach for improving enteric fever diagnostics by detecting the molecular host immune response patterns occurring during invasive *Salmonella* infection. To achieve this, we used large datasets of host gene expression profiles representing confirmed enteric fever cases and other causes of undifferentiated febrile illnesses. Machine learning analysis identified five genes for which differential activity could identify the enteric fever cases with an accuracy of over 96%, further validated in independent patient cohorts.

**Impact**
This cutting-edge, data-driven approach utilizes the increasing amount of molecular immunology data accumulating in the public domain and combines advanced analytics with biology and global health. Using this type of molecular signature may significantly improve the detection and management of enteric fever and other causes of undifferentiated febrile illness.

accuracies (Brodersen *et al*, 2010) were recorded to account for class imbalances. Genes were then ranked by the frequency of positive gene selection by GRRF (based on mean-Gini) during the 100 iterations, and only genes included in at least 25% of the selection rounds were included in the diagnostic signature and used for prediction of the independent validation cohort as well as the samples belonging to the unknown cohort (Fig 3).

**High-throughput qPCR validation**

We performed TaqMan gene expression assays to validate gene expression levels in samples from Nepal and a subset of individuals from the Oxford challenge studies. A panel of 6 probes was measured in triplicates on a 192.24 Fluidigm chip using the Biomark at the Weatherall Institute for Molecular Medicine (WIMM) single-cell facility. Four samples and one probe failed in the quality control and were removed from the analysis. Raw Ct values were normalized to the housekeeping gene cyclophilin A (PPIA; $\Delta C_t$ values) and subsequently to control samples (healthy controls) to achieve $\Delta\Delta C_t$ values. The following primers were used: STAT1 (Hs01013996_m1), SLAMF8 (Hs00975302_g1), PSME2 (Hs01581610_g1), WARS (Hs00188259_m1), ALDH1A1 (Hs00946916_m1) and the housekeeping gene PPIA (Hs04194521_s1).

**Statistical analysis**

All data were processed in R version 3.2.4. Comparison of groups in Fig 3D was performed using Student's *t*-test (alternative: two-sided), correlations between clinical parameters and expression scores were performed using the Pearson correlation, and correlation between array and qPCR expression was performed using the Spearman correlations (alternative: two-sided).

## Data availability

The datasets produced in the present study are available in the Gene Expression Omnibus (GEO) database under the identifier GSE113867 (https://www.ncbi.nlm.nih.gov/geo/query/acc.cgi?acc= GSE113867).

**Expanded View** for this article is available online.

## Acknowledgements

We gratefully acknowledge the assistance of the participants who have taken part in the study both in Oxford and in Nepal. We are also grateful for the support from Laura B. Martin and GSK Vaccines for Global Health for setting up the paratyphoid challenge model and providing the *S.* Paratyphi O:2 antigen and the *S.* Paratyphi A challenge strain (NVGH308). We are grateful for the support from Myron M. Levine for providing the *S.* Typhi Quailes strain used in the typhoid CHIM. This work was supported by the Bill and Melinda Gates Foundation (OPP1089317 and OPP1084259); funding for the challenge studies was provided by a Wellcome Trust Strategic Translational Award (grant number 092661 to AJP); the European Vaccine Initiative (ref: PIM); European Commission FP7 grant "Advanced Immunization Technologies" (ADITEC); and the NIHR Oxford Biomedical Research Centre (Clinical Research Fellowship to TCD; oxfordbrc.nihr.ac.uk). For the support of the Fluidigm experiment, we acknowledge the Weatherall Institute of Molecular Medicine (WIMM) single-cell facility, which was supported by the MRC-funded Oxford Consortium for Single-cell Biology (MR/M00919X/1) and the Oxford-Wellcome Trust Institutional Strategic Support Fund. TCD, CJ, CJB, CSW and AJP are supported by the NIHR Oxford Biomedical Research Centre (Oxford University Hospitals NHS Trust, Oxford), TCD is NIHR-funded Academic Clinical Lecturer; SB is Sir Henry Dale Fellow, jointly funded by the Wellcome Trust and the Royal Society (100087/Z/12/Z); AJP is Jenner Investigator, James Martin Senior Fellow and NIHR Senior Investigator. The views expressed in this article are those of the author(s) and not necessarily those of the NHS, the NIHR or the Department of Health.

## Author contributions

All analyses were designed and performed by JM and CJB. The O3NP study was designed by TCD, AJP, SB, CJB and BB. Samples were collected by SD and AK. Challenge studies were designed and performed by MMG, HD, SS, CJ, TCD, CJB and AJP. RNA extractions and gene expression data generation were performed by SS, SP, HH, LB, FS, DP, TCD, CJB and GD. Graphical representation was created by JM and CJB.

## Conflict of interest

AJP chairs the UK Department of Health and Social Care's (DHCSC) Joint Committee on Vaccination and Immunisation and the EMA Scientific Advisory Group on vaccines, and he is a member of WHO's Strategic Advisory Group of Experts. The views expressed in the publication are those of the author(s) and not necessarily those of the DHSC, NIHR or WHO.

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
