## [Review Process File · EMBO Molecular Medicine]

Diagnostic host gene signature for distinguishing enteric fever from other febrile diseases

C.J. Blohmke, J. Müller, M.M. Gibani, H. Dobinson, S. Shrestha, S. Perinparajah, C. Jin, H. Hughes, L. Blackwell, S. Dongol, A. Karkey, F. Schreiber, D. Pickard, B. Basnyat, G. Dougan, S. Baker, A.J. Pollard, T.C. Darton

Review timeline:

Submission date:	24 February 2019
Editorial Decision:	26 March 2019
Revision received:	2 July 2019
Editorial Decision:	16 July 2019
Revision received:	30 July 2019
Accepted:	9 August 2019

Editor: Céline Carret

Transaction Report:

1st Editorial Decision

26 March 2019

Thank you for the submission of your manuscript to EMBO Molecular Medicine. We have now heard back from the three referees whom we asked to evaluate your manuscript.

You will see from the set of comments pasted below that all referees are enthusiastic about the findings. As such we would like to invite a revision of this work. We would like to encourage you to state the limitations of your work more clearly (see referees #1, #2 and #3) and address some items highlighted by referees #1 and #2. Of importance, an item that came out after cross-commenting is about including other bacteria in your study and more details/clarification of the findings. Depending on the nature of the revisions, this may be sent back to the referees for another round of review.

Please note that EMBO Molecular Medicine strongly supports a single round of revision and that, as acceptance or rejection of the manuscript will depend on another round of review, your responses should be as complete as possible.

Please read below for important editorial requests and consult our author's guidelines for proper formatting of your revised article for EMBO Molecular Medicine.

I look forward to receiving your revised manuscript.

***** Reviewer's comments *****

Referee #1 (Remarks for Author):

The manuscript titled "Diagnostic host gene signature to accurately distinguish enteric fever from other febrile diseases" by Blohmke et al describes identification of host response genes that could be used to discriminate enteric fever from other febrile illnesses. This is a well written manuscript with a lot of interesting data. I have a few major comments and a number of minor comments which should be addressed to make this paper suitable for publication.

Major comments

1. Overall, the authors overstate the potential utility of the diagnostic signatures identified in this study and some of the language should be toned down. In particular, it is an overstatement to suggest that this diagnostic gene signature can accurately distinguish enteric fever from other febrile illnesses. This suggests that the diagnostic has been evaluated in the field. Please remove "accurately". Similar comment for the one sentence summary. Please remove "reliably". Although the data is very promising there is a lot of additional work that needs to be done. Please include some discussion of the limitations of the findings or applicability and what else needs to be further tested (generalizability) of these findings.
2. Are culture-negative suspected enteric fever cases negative for any bacteria or Salmonella? If negative for all bacteria, why were samples from other febrile illnesses (e.g., pneumococcus) not included in the validation? In some areas where typhoid is endemic, febrile illnesses due to other bacteria are very common.
3. I found it very difficult to follow the groups/samples tested. The terminology/group names were not consistent throughout. See my specific comments below. Please ensure that the terminology is consistent and that groups are clear.

Minor comments:

1. Line 34. Define AUROC
2. Line 48. Should be "sensitivity"
3. Line 52. Change to "These reports highlight".
4. Line 53. Change to "to guide management of febrile patients and appropriate"
5. Line 70. Change "or" to "of"?
6. Line 74. Define EF.
7. Line 86 and throughout. Should be "S. Paratyphi A" instead of "S. Paratyphi".
8. Line 87. Are the culture-negative suspected enteric fever cases negative for any bacteria or Salmonella?
9. Line 97. Define BH.
10. Line 101. Why 69? Remove s from red squares, there is only 1 red square. Why is OXF Paratyphoid also included in the square? Shouldn't it be Nepal Typhoid and Nepal Paratyphoid?
11. Line 105 - move period after (Figure 1C-e) and add it after reference (14)
12. Line 122-123. Group 1 is labeled as Group A in Fig 1g. Group 2 is labeled as Group B and Group 3 is labeled as Group C.
13. Line 133. Were these febrile patients tested for any other infectious agents?
14. Line 135. Define RCT.
15. Line 135-138. Recent RCT performed in Nepal? Reference? Difficult to distinguish between which infections? Murine and scrub typhus or between typhus and enteric fever? This sentence is hypothetical and should be moved to the discussion.
16. Line 150, Remove one of the "febrile"
17. Line 152. Develop?
18. Line 167. Define AUROC
19. Line 169. Why >0.5 ? How was this chosen?
20. Line 215. Does Figure 4c show microarray data or qPCR? Figure legend does not match the text.
21. Line 224. How is the expression score generated?
22. Line 227. Add "a" to "showed a poor correlation"
23. There is no reference to Supplementary Figure 5 in the text.
24. Line 242. Which data does the accuracy $>96\%$ refer to? Normally diagnostics are evaluated according to their sensitivity, specificity, negative predictive value and positive predictive value.
25. Line 242-244. I don't understand this sentence. This is how a diagnostic would be used. I.e., using samples collected directly from patient or participants. Also, this is not a fair comparison. You

are comparing sensitivity/specificity of your assay using data generated in a single laboratory on archived samples versus sensitivity/specificity of diagnostic methods evaluated in the field. Also, you haven't shown how well this gene signature can discriminate enteric fever from other bacterial febrile illnesses (e.g., pneumococcus, H. influenzae, iNTS, Klebsiella etc) that are also common in the same populations as enteric fever.

26. Line 250. Are you providing a reference to a diagnostic that could be adapted to measure the gene signature that you have identified? Not clear.
27. Line 255. Define DE (if not defined already).
28. Line 274. Add "a" to "SLAMF8 is a surface-"
29. Line 275. Found "on" macrophages?
30. Line 277. Add period at end of sentence.
31. Lines 357, 363, 364, 368, 382, 390, 395, 398, 399. Leave a space between the number and the unit.
32. Line 378. Negative for any pathogen or Salmonella?
33. Line 390. Onto bead-arrays?
34. Line 401. Is featureCounts a program?
35. Line 443. What is mtry?
36. Line 446. Suggest that you change "held out" to "remaining".
37. Line 474. What is ref: PIM?
38. Line 491. Should be no competing interests.
39. References. Bacterial names should be italicized.
40. Line 725. Up and down regulated compared to healthy community controls?
41. Line 747. Add "a" to "detect a 5-gene"
42. Line 763 and 765. ddCt?
43. Line 764. RF defined? Random Forest?
44. Line 778. What do blue and red represent? Expression compared to?
45. Figure 1. What is the 03NP-CONT group? Not shown in Fig 1a....
46. Figure 2. The groups don't match what is shown in Suppl. Table S2. E.g., what is GSE34404? Legend. Change to "Simultaneous differentiation"
47. Figure 3a. What do the grey bars represent? What do the green datapoints represent? What is the red datapoint?
48. Figure 3c. What is the unit?
49. Figure 3d. ST is from CHIM? Should this be T2-ST? SPT is P1-SPT?
50. Figure 4f. TD is T2-ST?
51. Figure 4g. What studies are the samples from? Nepal? CHIM?
52. Figure S1. Add key for colours to suppl. Fig legend. Note: the colours weren't obvious on my printer.
53. Figure S3. Same comments as for Figure 3.

Referee #2 (Remarks for Author):

This manuscript reports development of a 5 host gene signature that has high sensitivity and specificity for the diagnosis of enteric fever.

The strengths of the paper include: 1) it addresses a persistent and substantial clinical diagnostic problem; 2) it uses robust experimental design and statistical approaches in pursuit of the goal of a host gene panel for diagnostic utility; 3) the design includes both well-controlled human experimental subjects and real world samples; 4) the candidate panel and approach is tested against other diseases that are commonly in the differential diagnosis of enteric fever; 5) the results provide an encouraging first step towards development of a practical clinical test.

The weaknesses of the paper are minor and include:

1. A more complete picture of the differential diagnosis challenge for the putative assay would be useful, i.e. what are the top 5 top 10 diagnoses when enteric fever is considered but not present? In other words which infections/diseases should the panel be able to discriminate besides the dengue/malaria/tb which were (commendably) already included? If the answer is that there are no other 'suspects', this should be explicitly stated. If there are other enteritidies that are common and need to be distinguished from enteric fever, this should also be explicitly stated and discussed.

2. The introduction mentions the diagnostic confounding by patients who may have received some (inappropriate?) antibiotics prior to full evaluation. If pre-treated patients represent a significant fraction of suspected enteric fever cases, then the panel needs to be tested for efficacy in people whose host gene expression may be modulated by such treatment. This will be/is difficult but should be at least discussed as a limitation or potential factor for refining the assay.

3. The discussion is missing consideration of the practical obstacles to implementing a PCR based assay in the clinics and countries where this problem is found. The authors should offer an opinion on whether current rapid PCR platforms can be successfully adapted to these locations, especially in terms of cost. One assumes they think the answer is yes, but some reasons for this position would benefit the reader.

Some additional minor points:

1. The numbers for each group in this text should be explained. "We designed a

146 discovery cohort consisting of control samples from each respective study (n=220 community controls or convalescent samples, 'CTRL'), 74 enteric fever ('EF'), 94 blood stage *P. falciparum* ('bsPf'), 67 dengue ('DENV') and 54 active pulmonary tuberculosis ('PTB') cases. An independent validation cohort consisted of 109 CTRLs, 50 EF, 19 bsPf, 49 DENV, and 97 PTB samples ..." I think the answer to how these numbers were chosen are implied in the methods section of how different studies were divided into discovery vs validation, but an explicitly statement would be helpful that explains how these numbers were chosen.

2. typo: not elop enteric fever

3. I think the explanation/key to the modules of gene expression is implied to be offered in a prior publication (ref 14). If this is the case, it should be stated explicitly. If not, the modules should be defined and explained in an appendix here.

4. personal preference: The circular plot in 1C is a format that is au courant but I suspect few readers actually are able to extract useful information from this dense plot, and its message is described only briefly in the results and figure legend. My difficulty with this graph style may be representative of only a minority of readers, but for this minority a more expanded sentence or two of what the authors want the reader to learn from this figure would be helpful.

Referee #3 (Remarks for Author):

I can only comment generally on this MS as I have no experience in 'omics technology. Overall the approach used here is interesting and the use of these types of data sets for diagnosis, particularly of diseases where it is difficult to distinguish the underlying cause such as febrile diseases, is clearly the way forward for the future. My question on the MS as presented is that the authors do not appear to achieve clear diagnostic signatures as yet, although there are some very interesting data presented. The authors do get part of the way to a diagnostic signature and maybe this is the best that can be achieved, but if these data were validated against more patients then it may be possible to obtain a series of clear diagnostic patterns in the gene signatures.. The paper is very close to having a clear story (negative or positive) and it's currently left a little incomplete. If the approach is not going to bring diagnostic clarity then the data become a useful resource and it is important that this outcome is published. If it is going to work with more patients then this is very important and also needs to be published.

1st Revision - authors' response

2 July 2019

Referee #1 (Remarks for Author):

The manuscript titled "Diagnostic host gene signature to accurately distinguish enteric fever from other febrile diseases" by Blohmke et al describes identification of host response genes that could be used to discriminate enteric fever from other febrile illnesses. This is a well written manuscript with a lot of interesting data. I have a few major comments and a number of minor comments which

should be addressed to make this paper suitable for publication.

Major comments

1. Overall, the authors overstate the potential utility of the diagnostic signatures identified in this study and some of the language should be toned down. In particular, it is an overstatement to suggest that this diagnostic gene signature can accurately distinguish enteric fever from other febrile illnesses. This suggests that the diagnostic has been evaluated in the field. Please remove "accurately". Similar comment for the one sentence summary. Please remove "reliably". Although the data is very promising there is a lot of additional work that needs to be done. Please include some discussion of the limitations of the findings or applicability and what else needs to be further tested (generalizability) of these findings.

We thank the reviewer for this insightful comment. First, we have removed "accurately" from the following sections:

- "Accurately" Removed from:
 - Title. Line 1. And changed to "*Diagnostic host gene signature **for distinguishing enteric fever from other febrile diseases.***"
 - Abstract. Line 34.
 - Introduction. Line 76 revised manuscript.
 - Results. Line 186 revised manuscript. And changed to "*...**for predicting...***"
 - Results subheading. Line 189 revised manuscript.
 - Results. Line 197 revised manuscript.

And to more accurately convey the important message from our manuscript and to remove the overstatement of our results, the summary sentence has been changed from:

Using host gene expression to reliably diagnose culture confirmed enteric fever cases from other subjects with febrile illnesses.

To ('reliably' removed):

"Signatures derived from host gene expression responses demonstrate potential for distinguishing culture-confirmed enteric fever cases from other causes of febrile illness in challenge study and an endemic setting."

We further agree with the reviewer's comments that the data are promising but that further work needs to be required prior to delivering a product for use in endemic settings. We recognise this by mentioning that further optimisation is required (line 248 revised manuscript) and feel that demonstration of detecting the upregulated genes using a qPCR method goes some way towards demonstrating how this may be technically feasible. Limitations to our analysis that are discussed include the paucity of publicly available datasets (lines 329 onwards revised manuscript), in particular as relate to key pathogens such as rickettsial infection.

We are currently setting up a large multisite prospective diagnostic study in order to address shortcoming including small samples sizes, lack of gene expression data and missing fever causing aetiologies (e.g. rickettsia) (see response to comment 2 below). To further acknowledge the study's limitations, we included the following narrative in lines 348-355 (revised manuscript):

"Despite all this evidence, an open question remains the effect of prior antibiotic use and duration of illness on the expression of the 5-gene signature. While our study is too small to formally assess this, the similarity of gene expression between samples derived from the CHIM (short disease duration, no antibiotic exposure) and Nepal (long disease duration and exposure to antibiotics) may suggest stability of the gene signature in the presence of these co-founding factors. Additional work is required to address these limitations by performing a multisite prospective diagnostic evaluation recruiting all patients with fever to ensure a balanced case mix and range of alternate fever causing aetiologies (e.g. rickettsial infection)."

2. Are culture-negative suspected enteric fever cases negative for any bacteria or Salmonella? If negative for all bacteria, why were samples from other febrile illnesses (e.g., pneumococcus) not included in the validation? In some areas where typhoid is endemic, febrile illnesses due to other bacteria are very common.

In endemic settings, most clinicians and research studies use the WHO definition for making a diagnosis of suspected enteric fever which includes fever for 72 hours or more (1) generally in the absence of other foci of infection. This describes the presenting characteristics of participants who were recruited to the Nepal endemic cohort. Among these, the vast majority (>90%) who were culture positive had *S. Typhi* or *S. Paratyphi A* isolated from blood. The addition of prolonged fever duration to no clinically obvious focus of infection essentially excludes those patients who have other bacterial infections including *S. pneumoniae*. Other probable causes of fever in this group of patients (based on our previous data and experience in this setting)(2) may include Rickettsial infection (murine and tick typhus), hantavirus and Q fever, and probably leptospirosis. Diagnosis of these infections in any setting is complex and generally requires national or international reference laboratory support, which were not available in the context of this study.

A major hurdle in including these disease aetiologies in the types of gene expression analysis presented is the (current) lack of availability of deposited gene expression datasets for these other types of infection. To develop and perform advanced analytics including machine-learning approaches, sufficient data are currently available only for dengue, malaria and TB. While there may be differences in clinical presentation – i.e. they usually present as acute fever and are often diagnosed within an hour of presentation due to the availability of rapid tests (dengue and malaria) or present with longer term chronic respiratory and or constitutional symptoms – we took the pragmatic approach that these are still relevant to the population being investigated for undifferentiated febrile illness in Nepal.

To address the reviewer’s comment, we added “...with ≥ 3 days of fever, with no obvious focus of infection...” to the methods section (line 397-398 revised manuscript) to add more detail regarding the criteria of febrile patients enrolled in the Nepalese cohort. In addition we have added the following text to the discussion to highlight the limitation of data availability, line 329-334 (revised manuscript): ***“Indirect evidence for a likely high burden of rickettsial infection was demonstrated in a recent randomised controlled treatment trial of treatment for typhoid at the same centre in Nepal (3). In this study a higher proportion of culture-negative cases clinically responded to fluoroquinolone class antibiotics rather than the 3rd generation cephalosporin used in the study, probably due to the high frequency of murine and scrub typhus presenting as acute undifferentiated febrile infection in this population.”***

3. I found it very difficult to follow the groups/samples tested. The terminology/group names were not consistent throughout. See my specific comments below. Please ensure that the terminology is consistent and that groups are clear.

Indeed, in a complex study like ours this is important. We thoroughly went through the manuscript and clarified the terminology and group labels. We amended the text at the following positions by adding the relevant study specific sample name (where possible) in line 119, 123, 124, 127, 129, 208, 212, 216, 217 and the legend of figure 4 (revised manuscript).

In addition, we changed:

- The track label of Figure 1C so it reflects the study, not just the sample, in line with figure 1A and the figure legend: “nD7 Paratyphoid (OXF)” and “nD7 Typhoid (OXF)” now reads “***P1-nD7***” and “***T1-nD7***”, respectively.
- The x-axis of Figure 4c “CTRL. 03NP-sEF ST SPT” now reads “CTRL. 03NP-sEF ***03NP-ST 03NP-SPT***” and the legend for Figure 4C accordingly.
- Colour legend of Figure 4e and x-axis Figure 4f: ST and SPT now read ***03NP-ST*** and ***03NP-SPT***, respectively. And we amended the figure legend accordingly.
- Figure S4: In Supplementary Figure S4 panel A represents the samples from the Discovery cohort and panel B the samples from the validation cohort. We ensured that this is highlighted in the figure legend.

Minor comments:

1. Line 34. Define AUROC

Abstract (line 35): Changed to: “(area under receiver operating characteristic curve >95%)”

2. Line 48. Should be "sensitivity".

Changed to: “... sensitivity...” (line 50 revised manuscript).

3. Line 52. Change to "These reports highlight".
This was changed accordingly (line 54 revised manuscript).
4. Line 53. Change to "to guide management of febrile patients and appropriate"
Changed to: "... *to guide management of febrile patients and appropriate* ..." (line 55 revised manuscript).
5. Line 70. Change "or" to "of"?
Changed to: "...*of*..." (Line 72 revised manuscript).
6. Line 74. Define EF.
Changed to: "... *enteric fever (EF)* ..." (line 76 revised manuscript).
7. Line 86 and throughout. Should be "S. Paratyphi A" instead of "S. Paratyphi".
Changed as suggested, in lines (revised manuscript) 88, 92, 95, 123, 129, 226, 257, 259, 499, and legend of Figure 1-4.
8. Line 87. Are the culture-negative suspected enteric fever cases negative for any bacteria or Salmonella?
Please refer to response to major comment 2 above.
9. Line 97. Define BH.
Changed to: "... (*Benjamini-Hochberg (BH) corrected* ..." (line 99 revised manuscript).
10. Line 101. Why 69? Remove s from red squares, there is only 1 red square. Why is OXF Paratyphoid also included in the square? Shouldn't it be Nepal Typhoid and Nepal Paratyphoid? These numbers represent the percentages of overlap of significantly enriched expression modules (BTMs) during *S. Typhi* or *S. Paratyphi A* infection between the Nepal and Oxford cohorts. Thus, for *S. Typhi* 56% of BTMs overlap between Nepal and the Oxford Challenge model. For *S. Paratyphi A* 69% of BTMs overlap between Nepal and the Oxford Challenge model. These numbers (a '%' sign was missing in the original narrative) are reflected in the table and mentioned in the text in *line 103* (revised manuscript). Instead of red squares in Supplementary Table 1 we highlighted the relevant cells in Table S1. We hope this clarifies this table and the numbers in the text.
11. Line 105 - move period after (Figure 1C-e) and add it after reference (14).
Changed as suggested.
12. Line 122-123. Group 1 is labelled as Group A in Fig 1g. Group 2 is labelled as Group B and Group 3 is labelled as Group C.
Thank you for pointing this out, we have amended this accordingly, so the text is aligned with the figure (line 126-128 revised manuscript).
13. Line 133. Were these febrile patients tested for any other infectious agents?
Please see answer for major comment 2.
14. Line 135. Define RCT.
Please refer to answer to 15 below.
15. Line 135-138. Recent RCT performed in Nepal? Reference? Difficult to distinguish between which infections? Murine and scrub typhus or between typhus and enteric fever? This sentence is hypothetical and should be moved to the discussion.
As suggested the section "***These febrile patients were considered clinically to have enteric fever, and were therefore treated as such, however their heterogeneous gene transcription profiles suggest that any one of several different aetiologies may have precipitated hospital presentation. Further evidence to this is that in a recent RCT a higher proportion of culturenegative cases responded to fluoroquinolones rather than a 3rd generation cephalosporin, possibly due to the frequency of murine and scrub typhus in this population, however distinguishing between these infections is currently difficult.***" (this appeared at line 139 of the revised manuscript) has been removed and replaced with the following wording in the discussion section as suggested (329-334 revised manuscript):

“Indirect evidence for a likely high burden of rickettsial infection was demonstrated in a recent randomised controlled treatment trial of treatment for typhoid at the same centre in Nepal (3). In this study a higher proportion of culture-negative cases clinically responded to fluoroquinolone class antibiotics rather than the 3rd generation cephalosporin used in the study, probably due to the high frequency of murine and scrub typhus presenting as acute undifferentiated febrile infection in this population.”

16. Line 150, Remove one of the "febrile".

Line 152 revised manuscript: Changed to: "... ***febrile culture-negative suspected EF cases*** ..."

17. Line 152. Develop?

Line 153 revised manuscript: Changed from: "evlop" to "***develop***"

18. Line 167. Define AUROC

Line 168 revised manuscript: Changed from "(AUROC: 96.7%)" to "***(area under receiver operating characteristic curve, AUROC: 96.7%)***"

19. Line 169. Why >0.5? How was this chosen?

In a 2-class classification problem a prediction probability of 0.5 means that the probability is no better than chance (50:50) and thus is comparable to a coin toss – i.e. over time each class can be predicted with a 50% probability. Thus, typically the prediction probability must be above 0.5 to consider it not to be driven by chance. In figure 3c, the red line indicates the prediction probability of 0.5 (50%). We also clarified this in the figure legend 3c by adding: Figure legend 3c: "***... Red dotted line signifies the 0.5 prediction probability.***"

20. Line 215. Does Figure 4c show microarray data or qPCR? Figure legend does not match the text.

Figure 4 represents data from both technologies. Panel a&b are based on array data, where panel c, d&e are based on qPCR data. We added "***...(based on microarray data)...***" in line 215 (revised manuscript) and clarified this in the legend figure 4 a&b.

21. Line 224. How is the expression score generated?

The expression score is generated by taking the z-score of the geometric mean of the fluorescent intensity across the target genes (e.g. the 5 genes of the signature identified in this study) in each individual (see lines 173 revised manuscript).

22. Line 227. Add "a" to "showed a poor correlation"

We have amended this as requested (line 230 revised manuscript).

23. There is no reference to Supplementary Figure 5 in the text.

This is an oversight error and the reference has been included in line 211 (revised manuscript).

24. Line 242. Which data does the accuracy >96% refer to? Normally diagnostics are evaluated according to their sensitivity, specificity, negative predictive value and positive predictive value. This mention in the discussion (line 245 revised manuscript) refers to the related data in lines 166-169 (revised manuscript) reporting the results of predicting the validation cohort using the 5-gene signature in which the AUROC was found to be 96.7% with the sensitivity and specificity also reported there. We have included these data also in the discussion to clarify this point. We amended the sentence and hope this suffices as clarification as we are merely referring to the characteristics of the diagnostic signature.

Sentence now reads: "Here we demonstrate a reproducible host expression signature of 5 genes (*STAT1*, *SLAMF8*, *PSME2*, *WARS*, and *ALDH1A1*) able to discriminate EF cases from other common causes of fever in the tropics with an accuracy of >96% (AUROC; ***sensitivity 97%, specificity 88%***)." (line 243-246 revised manuscript).

25. Line 242-244. I don't understand this sentence. This is how a diagnostic would be used. I.e., using samples collected directly from patient or participants. Also, this is not a fair comparison. You are comparing sensitivity/specificity of your assay using data generated in a single laboratory on archived samples versus sensitivity/specificity of diagnostic methods evaluated in the field. Also, you haven't shown how well this gene signature can discriminate enteric fever from other bacterial

febrile illnesses (e.g., pneumococcus, H. influenzae, iNTS, Klebsiella etc) that are also common in the same populations as enteric fever.

We agree that this sentence is unclear and with the limitations to our results as suggested by the reviewer. It has therefore been removed (this sentence occurred originally in line 246 in the revised manuscript).

26. Line 250. Are you providing a reference to a diagnostic that could be adapted to measure the gene signature that you have identified? Not clear.

Our intention was to suggest that by confirming the presence of the gene signature identified using a different and more readily transferable method (i.e. qPCR), that might support its feasibility for development into a routine test rather than continued requirement for RNAseq or large gene array analysis. The sentence has been altered for clarity and overstatement and now reads (line 250 revised manuscript):

“While further optimisation work is required, validating the expression of our signature using conventional methods such as qPCR demonstrates feasibility *for* further development into a ***routine laboratory test (4)***.”

27. Line 255. Define DE (if not defined already).

This was first defined in the beginning of the results section, line 90 (revised manuscript).

28. Line 274. Add "a" to "SLAMF8 is a surface-"

Corrected as suggested (line 279 revised manuscript).

29. Line 275. Found "on" macrophages?

Corrected as suggested (line 280 revised manuscript).

30. Line 277. Add period at end of sentence.

Amended as suggested (line 282 revised manuscript).

31. Lines 357, 363, 364, 368, 382, 390, 395, 398, 399. Leave a space between the number and the unit.

Corrected as suggested.

32. Line 378. Negative for any pathogen or Salmonella?

Amended to read “...*febrile patients who were blood culture negative for any pathogen (n=71)*.” (line 399 revised manuscript).

33. Line 390. Onto bead-arrays?

Corrected as suggested (line 412 revised manuscript).

34. Line 401. Is featureCounts a program?

featureCounts is a function from the subread package (“subread set of tools”). We amended the sentence, so it now reads (line 424 revised manuscript) and added the appropriate online reference (5) to this R package:

“*featureCounts()*, a **function** from the subread set of tools v1.5.1 (62), was...”.

35. Line 443. What is mtry?

mtry is a parameter of the random forest algorithm which is determined by running a through a grid of values. It is a hypervariable typically used as part of the tuning process of machine learning algorithms in order to optimize the classification performance. Clearly this parameter is very specialized and not necessarily understandable to the average reader, however for transparency reasons we prefer leaving this in the manuscript in order to provide all information available for fellow computational scientists.

36. Line 446. Suggest that you change "held out" to "remaining".

Corrected as suggested (line 469 revised manuscript).

37. Line 474. What is ref: PIM?

PIM is the reference of the grant given by the European Vaccine Initiative (EVI).

38. Line 491. Should be no competing interests.

We have amended this to as per journal guidelines. It now reads (line 520-523 revised manuscript): ***“Conflict of interest: AJP chairs the UK Department of Health and Social Care’s (DHSC) Joint Committee on Vaccination and Immunisation and the EMA Scientific Advisory Group on vaccines, and he is a member of WHO’s Strategic Advisory Group of Experts. The views expressed in the publication are those of the author(s) and not necessarily those of the DHSC, NIHR or WHO.”***

39. References. Bacterial names should be italicized.

We have amended this accordingly.

40. Line 725. Up and down regulated compared to healthy community controls?

Line 774 (revised manuscript): We amended the sentence, so it now reads ***“Black numbers indicate the up- and down-regulated genes compared to healthy controls.”***

41. Line 747. Add "a" to "detect a 5-gene"

Corrected as suggested (line 799 revised manuscript).

42. Line 763 and 765. ddCt?

This is the delta delta Ct value typically calculated from reverse transcriptase quantitative PCR. It reflects the change from control samples and a comparator condition. We have changed this to reflect the symbol “ Δ ” rather than two “D’s” and this now reads (line 816, 819 revised manuscript): ***“... ($\Delta\Delta CT$ over PPIA) ...”***.

43. Line 764. RF defined? Random Forest?

Indeed, this means random forest and we spelled this out for clarification (line 818 revised manuscript).

44. Line 778. What do blue and red represent? Expression compared to?

For individual gene expression as well as modular enrichment this is compared to healthy control samples. We clarified this by adding in figure legend 1:

(Line 773 revised manuscript) ***“(b) Volcano plots of up (red) and down (blue) regulated genes (compared to healthy control samples) in...”*** and:

(Line 777 revised manuscript): ***“(c) ... BTM labels; direction of enrichment (blue: down; red: up; compared to healthy controls). ...”***.

45. Figure 1. What is the 03NP-CONT group? Not shown in Fig 1a....

We realized that these samples were not appropriately mentioned in the text. These blood culture samples were from three febrile individuals which yielded growth (two *Bacillus* sp. and one coagulase-negative *Staphylococcus*) which was interpreted as likely contamination and not associated with the clinical condition. They were therefore excluded from the remainder of the analysis. A sentence has been added to clarify this:

Line 129 revised manuscript: ***“Of note, three samples (‘03NP-CONT’) in this cohort grew bacterial contaminants and were thus removed from the entire analysis.”***

And mentioned this also in figure legend 1: ***“Three samples labelled as ‘03NP-CONT’ are samples that grew bacterial contaminants and were thus excluded from the entire analysis.”***

46. Figure 2. The groups don't match what is shown in Suppl. Table S2. E.g., what is GSE34404?

Legend. Change to "Simultaneous differentiation"

This is an oversight error and we have added a legend indicating what each study ID represents. Moreover, we have amended Table S2 to reflect this information, added a column signifying the study abbreviation used in Figure 2 and the text, and added which samples from each study have been used in each stage of the analysis (Discovery, Validation, Unknown cohort). Finally, to be clear in Figure 2, we indicated which dataset was included in each cohort. We expanded the legend to Table S2 as follows:

“...the discovery, validation, unknown and qPCR cohort. Column “Samples” signifies the samples of each dataset used in each part of the analysis (Discovery, Validation, Unknown cohort). D0 = Healthy, pre-challenge baseline samples from the challenge studies; CTRL = healthy community or convalescent controls; ST = S. Typhi; SPT = S. Paratyphi A; DENV = dengue virus samples; bsPf = blood-stage Plasmodium falciparum samples; PTB = pulmonary Tb samples; sEF =

febrile, culture-negative, suspected enteric fever samples; nD7 = day 7 samples derived from participants of the challenge study who stayed well the entire 14 day challenge period. Note: some studies repurposed from GEO include several datasets. For example, GSE37250 by Kaforou et al. contains two cohorts of tuberculosis patients – one cohort from South Africa and one cohort from Malawi.”

47. Figure 3a. What do the grey bars represent? What do the green datapoints represent? What is the red datapoint?

Figure 3a represents a box-and-whiskers plot (rotated 90 degrees). Because the GRRF algorithm is a regularized algorithm it has a build-in feature selection step. This means that in each of the 100 iterations of the 70/30 split (Discovery cohort), the algorithm selects a set of features that form the classifier (i.e. contribute to the successful split between EF and Rest). The selection is based on an importance measure which is a numerical value ranging from 1-100 (1 = most important; 100 = least important). In this figure, the genes (y-axis) are ranked by how often (out of 100 iterations) each gene was included in the classifier. The green dots indicate the importance measure each time a gene was chosen. Thus, for SLAMF8 it becomes clear that it was 98/100 times included in the classifier and always deemed very important (important measures <10). The orange dot represents the percentage of how often the gene was included in the classifier (SLAMF8 = 98%). For PSME2, WARS, and ALDH1A1 the box-and-whiskers plot appears like a grey bar because the genes have not always been included in the classifier (in fact only 39%, 37% and 36% respectively), hence the plot spans a wide range of importance measures (x-axis). We added the following information in Figure legend 3a:

“...of 100 iterations (orange dot) during the 2-class classification. Y-axis = genes ranked by selection frequency. X-axis = importance measure of each gene across all 100 iterations. Green dot: Importance measure for each gene per iteration. A cut-off ...”

We also amended figure 3a and 3e by adding arrows to the x-axis indicating that 1 = high importance, 100 = low importance.

48. Figure 3c. What is the unit?

The unit in Figure 3c is ‘Prediction Probability’. We amended Figure legend 3c so it now reads: **“...Y-axis: Prediction probability ranging from 0-1.”**

49. Figure 3d. ST is from CHIM? Should this be T2-ST? SPT is P1-SPT?

Because the discovery cohort included enteric fever samples representing *S. Typhi* and *S. Paratyphi A* samples from T1 (Oxford CHIM), P1 (Oxford CHIM) and 03NP (Nepal), in Figure 3d ST refers to *S. Typhi* and SPT to *S. Paratyphi A* from the CHIM as well as Nepal. However, in the bottom part (Validation cohort) of Figure 3d, ST samples refer to *S. Typhi* samples from the T2 CHIM study as per Table S2. We added a sentence in figure legend 3d to highlight this:

“... ST: S. Typhi. ST and SPT samples are derived from the challenge models as well as from Nepal.”

50. Figure 4f. TD is T2-ST?

TD here signifies ‘Typhoid Diagnosis’ samples from the CHIM. In fact, in this figure, the TD samples are neither from T1 or T2, but yet another independent cohort derived from a recently published vaccine study, which we explained in methods line 372-375 (revised manuscript). This is further mentioned in the results section (line 223 revised manuscript) where we refer to Table S2. This table has now also been amended to include the qPCR cohort and explaining which samples were used. To clarify this we added:

“... Table 2 – qPCR.” in line 223 (revised manuscript) before the reference to this independent study.

Moreover, we amended figure legend 4d to explain where the ‘TD’ samples were derived from by adding: *“...challenge (TD) in the Vi-TCV study (Table S2).”*

And finally, we added the clarification of each sample label in figure legend 4f:

“... .D0 = pre-challenge baseline Vi-TCV study; nD7 = day 7 samples of participants who stayed well following challenge (Vi-TCV study); SPT = S. Paratyphi A (03NP); ST = S. Typhi (03NP); TD = Typhoid diagnosis (Vi-TCV study).”

51. Figure 4g. What studies are the samples from? Nepal? CHIM?

Correlation of between expression and temperature was performed on samples from the independent CHIM (Vi-TCV study) and EF samples from Nepal (at presentation to hospital). CRP correlations

were only possible in CHIM samples as this marker was not measured in Nepal samples. We clarified this by changing the figure legend of figure 4g *from*:

“(g) Spearman’s rank correlation of the 5-gene combined expression score and temperature (left; only nD7 and TD samples from the Oxford CHIM and SPT and ST cases from Nepal were included) and CRP (right; CRP was only available for Oxford CHIM samples and we excluded D0 baseline measures) at presentation to hospital (Nepal), diagnosis (Oxford CHIM) or day 7 after challenge in those who stayed well (Oxford CHIM).”

To:

“(g) Spearman’s rank correlation of the 5-gene combined expression score and (**left**) temperature (only nD7 and TD samples from the Oxford CHIM – Vi-TCV and SPT and ST cases from Nepal at **presentation to hospital** were included) and (**right**) CRP (CRP was only available for Oxford CHIM – Vi-TCV samples and we excluded D0 baseline measures).”

52. Figure S1. Add key for colours to suppl. Fig legend. Note: the colours weren't obvious on my printer.

The colours of the dendrogram in the heatmap only correspond to the different clusters identified in the heatmap and have no further meaning, hence we left them as is. We indicated the colour labels for figure S1a in the legend by adding: “... (*blue: down-regulated; red: up-regulated compared to healthy community controls*).”

53. Figure S3. Same comments as for Figure 3.

We amended the legend for figure S3 in the same way as for figure 3a.

Referee #2 (Remarks for Author):

This manuscript reports development of a 5 host gene signature that has high sensitivity and specificity for the diagnosis of enteric fever.

The strengths of the paper include: 1) it addresses a persistent and substantial clinical diagnostic problem; 2) it uses robust experimental design and statistical approaches in pursuit of the goal of a host gene panel for diagnostic utility; 3) the design includes both well-controlled human experimental subjects and real world samples; 4) the candidate panel and approach is tested against other diseases that are commonly in the differential diagnosis of enteric fever; 5) the results provide an encouraging first step towards development of a practical clinical test.

The weaknesses of the paper are minor and include:

1. A more complete picture of the differential diagnosis challenge for the putative assay would be useful, i.e. what are the top 5 top 10 diagnoses when enteric fever is considered but not present? In other words which infections/diseases should the panel be able to discriminate besides the dengue/malaria/tb which were (commendably) already included? If the answer is that there are no other 'suspects', this should be explicitly stated. If there are other enteritidies that are common and need to be distinguished from enteric fever, this should also be explicitly stated and discussed.

We thank the reviewer for this comment and would like to refer to response to Reviewer 1 comment 2. In addition we have previously investigated the cause of non-specific febrile infection in the setting of Kathmandu (2) in which we identified some of the likely alternate diagnoses at least in this setting. As discussed above, there are currently no publicly accessible repositories containing gene expression data relevant to these other conditions which we could include in our analysis. There are no other common non-typhoidal *Salmonella* strains in circulation in this setting (Kathmandu) currently, as we have recently demonstrated: over 23 years, only 0.6% of bacteraemia has been caused by non-typhoidal *Salmonella* species (6). This has been added to line 334 (revised manuscript):

“Of note, we have shown that non-typhoidal *Salmonella* infection only accounts for 0.6% bacteraemia over a 23-year period (6).”

2. The introduction mentions the diagnostic confounding by patients who may have received some (inappropriate?) antibiotics prior to full evaluation. If pre-treated patients represent a significant fraction of suspected enteric fever cases, then the panel needs to be tested for efficacy in people whose host gene expression may be modulated by such treatment. This will be/is difficult but should

be at least discussed as a limitation or potential factor for refining the assay.

Indeed, several of the febrile participants in the Nepal cohort have received antibiotics prior to seeking care at the hospital, although these were few (approximately 31/102). We agree with the reviewer that antibiotic use prior to sampling may affect diagnostics in general, however we believe that this is more relevant to culture-based methods. This is indeed one of the motivations of seeking a different approach to diagnosis (mentioned in the introduction, line 47-50 revised manuscript). Our participant numbers are too small to perform a formal analysis of the effect of antibiotics on the gene signature and how it impacts sensitivity. A hint why this may be minimal is that in challenge model participants the strong gene signature present at diagnosis does persist at least 24hrs post commencement of antibiotic treatment (7). Moreover, the gene expression response of patients in Nepal (who have been febrile for at least 3 days and possibly been taking over the counter antibiotic treatment) is highly similar to participants diagnosed in the challenge model, providing further indication that in principle the overall response persists for the duration of the disease and possibly after antibiotic use. This suggests that duration of illness and 'uncontrolled' antibiotic use is unlikely to affect the signature. A formal study of the effect of antibiotics on gene expression would be a useful future exercise.

We acknowledge this as a limitation to the current study and have added the following text in the discussion:

Line 259 (revised manuscript) we added: *"...despite longer duration of disease and 'uncontrolled' exposure to antimicrobials in Nepal,..."*.

Line 348-355 (revised manuscript) we added: *"Despite all this evidence, an open question remains the effect of prior antibiotic use and duration of illness on the expression of the 5-gene signature. While our study is too small to formally assess this, the similarity of gene expression between samples derived from the CHIM (short disease duration, no antibiotic exposure) and Nepal (long disease duration and exposure to antibiotics) may suggest stability of the gene signature in the presence of these co-founding factors. Additional work is required to address these limitations by performing a multisite prospective diagnostic evaluation recruiting all patients with fever to ensure a balanced case mix and range of alternate fever causing aetiologies (e.g. rickettsial infection)."*

3. The discussion is missing consideration of the practical obstacles to implementing a PCR based assay in the clinics and countries where this problem is found. The authors should offer an opinion on whether current rapid PCR platforms can be successfully adapted to these locations, especially in terms of cost. One assumes they think the answer is yes, but some reasons for this position would benefit the reader.

It is difficult to estimate the costs for adapting a PCR test in resource limiting settings, as it is the early stages for such developments and further studies are needed to evaluate this. However, interesting developments in this area are underway, e.g. in developing highly sensitive amplification tests for water treatment (6) and mobile phone powered PCR technologies which would be a game changer in resource limited settings (3). Clearly, a concerted effort of biologist, medical doctors and engineers is required to evaluate whether such approaches are viable. While we don't feel to be in a position to adequately comment on cost estimates, we added a sentence to the discussion highlighting these developments (including reference (4, 8):

Line 250-254 (revised manuscript): *"A rapid, PCR-based test would be useful tool for accelerated diagnosis of enteric fever in hard to diagnose settings. While cost-effective, early approaches for easy use in resource-limited settings exist (25, 26), further development of reliable PCR diagnostic is needed to make such a test fit for purpose."*

Some additional minor points:

1. The numbers for each group in this text should be explained. "We designed a discovery cohort consisting of control samples from each respective study (n=220 community controls or convalescent samples, 'CTRL'), 74 enteric fever ('EF'), 94 blood stage *P. falciparum* ('bsPf'), 67 dengue ('DENV') and 54 active pulmonary tuberculosis ('PTB') cases. An independent validation cohort consisted of 109 CTRLs, 50 EF, 19 bsPf, 49 DENV, and 97 PTB samples ..." I think the answer to how these numbers were chosen are implied in the methods section of how different studies were divided into discovery vs validation, but an explicitly statement would be helpful that explains how these numbers were chosen.

We would like to clarify where these numbers came from: The cohorts were chosen based on a variety of motivations including the choice of samples from patients with undifferentiated febrile illnesses as well as availability of samples in the public domain. Each cohort has a varying number of samples and clinical presentations of the disease/infection of interest (e.g. tuberculosis cohorts contain pulmonary Tb samples, latent Tb samples and healthy controls). We aimed to include only clear sample definitions further varying the number of usable samples per cohort. The next decision-making point was to keep each cohort as a separate entity and not mix samples, as they were collected in different sites, for different studies. For example for malaria the discovery cohort contained samples from Malawi and the validation cohort malaria samples from Rwanda. While this may impose a conservative constrain on the analysis, the high prediction accuracy of the independent validation cohort indicates that the gene set identified is robust and can withstand sampling in different locations and patient groups. This of course leads to the fact that sample sets are ‘unbalanced’, which in some analysis approaches may lead to issues. However, we evaluated the prediction performance using the ‘balanced accuracy’ measure, which takes class imbalances into account. A problem with imbalanced group is that either sensitivity or specificity can be very low on the expense of the other; hence the balanced accuracy ($BA = (\text{sensitivity} + \text{specificity})/2$) takes this into account (5). We hope this ratifies the sample selection. To address this comment, we added the following text at line 146 (revised manuscript): “...*Using independent data sets, we designed...*”

2. typo: not elop enteric fever

Corrected as suggested (line 153 revised manuscript).

3. I think the explanation/key to the modules of gene expression is implied to be offered in a prior publication (ref 14). If this is the case, it should be stated explicitly. If not, the modules should be defined and explained in an appendix here.

As suggested, we added the reference in the legend to figure 1d&e.
“For further details on BTMs refer to reference (64).”

4. personal preference: The circular plot in 1C is a format that is au courant but I suspect few readers actually are able to extract useful information from this dense plot, and its message is described only briefly in the results and figure legend. My difficulty with this graph style may be representative of only a minority of readers, but for this minority a more expanded sentence or two of what the authors want the reader to learn from this figure would be helpful.

We thank the reviewer for this insight. Indeed, a challenge with multidimensional datasets is the graphical visualization and often these are not easy and require the attention of the reader. Since we here mean to illustrate the overlap of significantly expressed transcriptional modules (BTMs) and a multidimensional Venn-diagram is not practical, we opted for this circos plot. To highlight the information presented we amended the results section at the following locations:

Line 102 (revised manuscript): **“Figure 1c shows the enriched BTMs in each population with lines indicating when specific BTMs are also enriched in other populations (Figure 1c, please also refer to Supplementary Table 1).”**

Referee #3 (Remarks for Author):

I can only comment generally on this MS as I have no experience in 'omics technology. Overall the approach used here is interesting and the use of these types of data sets for diagnosis, particularly of diseases where it is difficult to distinguish the underlying cause such as febrile diseases, is clearly the way forward for the future. My question on the MS as presented is that the authors do not appear to achieve clear diagnostic signatures as yet, although there are some very interesting data presented. The authors do get part of the way to a diagnostic signature and maybe this is the best that can be achieved, but if these data were validated against more patients then it may be possible to obtain a series of clear diagnostic patterns in the gene signatures. The paper is very close to having a clear story (negative or positive) and it's currently left a little incomplete. If the approach is not going to bring diagnostic clarity then the data become a useful resource and it is important that this outcome is published. If it is going to work with more patients then this is very important and also needs to be published.

Many thanks for these supportive comments. We agree with the reviewer's summary of our key findings – what we have suggested here is one possible approach to breaking the deadlock in the development of diagnostic tests for febrile diseases in typhoid endemic settings. It is clear that further work is required, both to address the limitations raised by several of the reviewers (effect of prior antimicrobial exposure, the range of aetiologies with which febrile patients present and further multi-site prospective evaluation) with respect to our approach. Moreover, what is critically needed are 'big' data sets required to support the development of advanced learning-based computer algorithms for diagnostics and for the potential development of prognostic biomarkers and improved patient case management. We hope to have covered most of these limitations and thoughts in the responses to the other two reviewer's and would like to refer to those. Importantly, we would like to mention here that clearly validation of such signatures is paramount. Indeed, we are currently setting up a large prospective multisite site study sampling all febrile patients to increase sample sizes in a balanced mix of patients representing several, currently missing important fever causing aetiologies. This study will start soon and carry on for several years and will provide an invaluable dataset to validate the signature identified here.

1. W. H. Organization, Typhoid and other invasive salmonellosis. *WHO Vaccine-Preventable Disease Surveillance Standards*, 1-13 (2018).
2. C. N. Thompson, S. D. Blacksell, D. H. Paris, A. Arjyal, A. Karkey, S. Dongol, A. Giri, C. Dolecek, N. Day, S. Baker, G. Thwaites, J. Farrar, B. Basnyat, Undifferentiated febrile illness in Kathmandu, Nepal. *Am J Trop Med Hyg* **92**, 875-878 (2015).
3. A. Arjyal, B. Basnyat, H. T. Nhan, S. Koirala, A. Giri, N. Joshi, M. Shakya, K. R. Pathak, S. P. Mahat, S. P. Prajapati, N. Adhikari, R. Thapa, L. Merson, D. Gajurel, K. Lamsal, D. Lamsal, B. K. Yadav, G. Shah, P. Shrestha, S. Dongol, A. Karkey, C. N. Thompson, N. T. V. Thieu, D. P. Thanh, S. Baker, G. E. Thwaites, M. Wolbers, C. Dolecek, Gatifloxacin versus ceftriaxone for uncomplicated enteric fever in Nepal: an open-label, twocentre, randomised controlled trial. *Lancet Infect Dis* **16**, 535-545 (2016).
4. L. Jiang, M. Mancuso, Z. Lu, G. Akar, E. Cesarman, D. Erickson, Solar thermal polymerase chain reaction for smartphone-assisted molecular diagnostics. *Sci Rep* **4**, 4137 (2014).
5. K. H. Brodersen, C. S. Ong, K. E. Stephan, J. M. Buhmann, The balanced accuracy and its posterior distribution. *2010 20th International Conference on Pattern Recognition (ICPR)*, 3121-3124 (2010).
6. R. M. Zellweger, B. Basnyat, P. Shrestha, K. G. Prajapati, S. Dongol, P. K. Sharma, S. Koirala, T. C. Darton, C. Boinett, C. N. Thompson, G. E. Thwaites, S. Baker, A. Karkey, Changing Antimicrobial Resistance Trends in Kathmandu, Nepal: A 23-Year Retrospective Analysis of Bacteraemia. *Front Med (Lausanne)* **5**, 262 (2018).
7. C. J. Blohmke, T. C. Darton, C. Jones, N. M. Suarez, C. S. Waddington, B. Angus, L. Zhou, J. Hill, S. Clare, L. Kane, S. Mukhopadhyay, F. Schreiber, M. A. Duque-Correa, J. C. Wright, T. I. Roumeliotis, L. Yu, J. S. Choudhary, A. Mejias, O. Ramilo, M. Shanyinde, M. B. Szein, R. A. Kingsley, S. Lockhart, M. M. Levine, D. J. Lynn, G. Dougan, A. J. Pollard, Interferon-driven alterations of the host's amino acid metabolism in the pathogenesis of typhoid fever. *J Exp Med* **213**, 1061-1077 (2016).
8. X. Huang, X. Lin, K. Urmann, L. Li, X. Xie, S. Jiang, M. R. Hoffmann, Smartphone-Based in-Gel Loop-Mediated Isothermal Amplification (gLAMP) System Enables Rapid Coliphage MS2 Quantification in Environmental Waters. *Environ Sci Technol* **52**, 6399-6407 (2018).

2nd Editorial Decision

16 July 2019

Thank you for the submission of your revised manuscript to EMBO Molecular Medicine. We have now received the enclosed reports from the referees that were asked to re-assess it. As you will see the reviewers are now supportive and I am pleased to inform you that we will be able to accept your manuscript pending minor editorial amendments.

***** Reviewer's comments *****

Referee #1 (Remarks for Author):

The article is suitable for publication.

Referee #2 (Remarks for Author):

The revisions have substantially improved the paper.

Referee #3 (Remarks for Author):

The authors have fully addressed my comments

2nd Revision - authors' response

30 July 2019

Authors made the requested editorial changes.

Corresponding Author Name: Christoph Blohmke

EMBO Molecular Medicine

Manuscript Number: EMM-2019-10431